# TRPML1 suppresses pulmonary fibrosis by limiting collagen and elastin deposition

Eva-Maria Weiden [1,14], Zala Serianz [1,14], Yvonne Klingl [2,14], Simone Jörs [3,14], Dawid Jaślan [1], Marco Keller [4], Sandra Prat Castro[1], Mane Mkhitaryan [1], Aicha Jeridi[5], Daria Briukhovetska [6], Barbara Spix[1], Anna Scotto Rosato[1], Ahmed Agami[5], Herbert B Schiller[5], Suhasini Rajan[1], Johann Schredelseker [1], Giorgio Fois [7], Manfred Frick[7], Sebastian Kobold[6,8,9], Margarethe Klein [10], Fabian Geisler[3], Jorge Garcia-Fortanet [11], Leon O Murphy[11], Franz Bracher [4], Christian Wahl-Schott[12], Thomas Gudermann [1,5], Alexander Dietrich [1,5], Martin Biel [4✉], Ali Önder Yildirim [5✉] & Christian Grimm [1,2,13✉]

## Abstract

In pulmonary fibrosis lung tissue is thickened and scarred, and the lungs become progressively stiffer and smaller, leading to low levels of blood oxygen and shortness of breath. Lung fibrosis is not curable and life expectancy is reduced. Fibrosis is characterized by an increased accumulation of extracellular matrix (ECM) proteins such as collagen and elastin. ECM proteins are degraded predominantly by matrix metalloproteinases (MMPs). Here, we show that the lysosomal cation channel TRPML1, which causes the lysosomal storage disorder mucolipidosis type IV (MLIV) when mutated or lost, regulates the levels of MMPs in the ECM of mouse airways, modulating exocytosis of MMP2, 8, 9, 12, and 19, which mediate collagen/elastin degradation. While TRPML1 loss reduces MMP levels in lung macrophage and fibroblast supernatants, small molecule activation of TRPML1 results in increased levels. MLIV mice display a fibrosis-like lung phenotype similar to the phenotype evoked by bleomycin. We thus identify TRPML1 as a regulator of MMP release in the lung with loss of TRPML1 resulting in lung fibrosis due to excessive extracellular collagen and elastin accumulation.

**Keywords** TRPML; TRPML1; TRPML3; Mcoln1; Pulmonary Fibrosis
**Subject Categories** Membranes & Trafficking; Molecular Biology of Disease; Respiratory System

## Introduction

Lung or pulmonary fibrosis (PF) is associated with high morbidity and mortality. Currently available therapies are not curative, with the exception of lung transplantation, and are not effective in reducing patient mortality. New therapeutic options are therefore urgently needed. One characteristic feature of PF is an increased deposition/reduced degradation of extracellular matrix (ECM), and the balance between ECM proteases, in particular matrix metalloproteinases (MMPs) and inhibitors thereof (e.g., TIMPs) is disturbed. As a result, an excessive build-up of fibrillary collagens and other ECM components occurs. Fibroblasts, including myofibroblasts, alveolar epithelial cells, and macrophages, are all involved in this process; however, specifically fibroblasts and macrophages are the major regulators of ECM degradation (Duarte et al, 2015; Zhao et al, 2022). MMPs are best known for their role in degrading ECM components such as collagen, elastin, or fibronectin. MMPs are trafficked intracellularly and can be stored in and released via intracellular vesicles, including endolysosomes. Their exocytosis can occur through exocytic vesicles, lysosomes, exosome release by multivesicular bodies (MVBs), or budding from the plasma membrane, while their endocytosis involves early endosomes (EE) (Hey et al, 2022). MMPs may also be degraded in lysosomes or reshuffled back to the plasma membrane through recycling endosomes (Hey et al, 2022; Chuliá-Peris et al, 2022; Buratta et al, 2020). MMPs that are formally classified as collagenases include MMP1 (collagenase 1), MMP8 (collagenase 2), and MMP13 (collagenase 3), but also other MMPs play important roles in collagen degradation such as MMP2, MMP9, MMP12, MMP14 (also known as MT1-MMP), or MMP19 (Balbín et al, 2001; Taddese et al, 2010; McGarry Houghton, 2015; Yun

[1]Walther Straub Institute of Pharmacology and Toxicology, Faculty of Medicine, Ludwig-Maximilians-University, Munich, Germany. [2]Immunology, Infection and Pandemic Research IIP, Fraunhofer Institute for Translational Medicine and Pharmacology ITMP, Munich/Frankfurt, Germany. [3]TUM School of Medicine and Health, Department of Clinical Medicine – Clinical Department for Internal Medicine II, University Medical Center, Technical University of Munich, Munich, Germany. [4]Department of Pharmacy, Ludwig-Maximilians-University, Munich, Germany. [5]Comprehensive Pneumology Center, Institute of Lung Biology and Disease, Helmholtz Zentrum München, Member of the German Center for Lung Research (DZL), Munich, Germany. [6]Division of Clinical Pharmacology, Department of Medicine IV, University Hospital Munich, Munich, Germany. [7]Department of Physiology, University of Ulm, Ulm, Germany. [8]German Cancer Consortium (DKTK), partner site Munich, Munich, Germany. [9]German Center for Lung Research (DZL), partner site Munich, Munich, Germany. [10]Institute for Neurophysiology, Hannover Medical School, Hannover, Germany. [11]Casma Therapeutics, 201 Brookline Ave, Boston, MA 02215, USA. [12]Institute of Cardiovascular Physiology and Pathophysiology, Ludwig-Maximilians-University, Munich, Germany. [13]Department of Pharmacology, Faculty of Medicine, University of Oxford, Oxford, UK. [14]These authors contributed equally: Eva-Maria Weiden, Zala Serianz, Yvonne Klingl, Simone Jörs.
✉E-mail: martin.biel@cup.uni-muenchen.de; oender.yildirim@helmholtz-muenchen.de; christian.grimm@med.uni-muenchen.de

et al, 2014). Thus, MMP2 is involved in the degradation of collagen I, II, III, IV, V, VII, and XI, MMP8 in the degradation of collagen I, II, III, V, VII, VIII, and X, MMP9 (collagenase 4) in the degradation of collagen I, III, IV, V, X, XI and XIV, while MMP12 is involved in degradation of fibronectin, elastin and partially also collagens such as collagen I, III, and IV (Taddese et al, 2010; Yun et al, 2014; Sand et al, 2013; Niu et al, 2016). MMP14 KO mice were shown to accumulate extracellular collagen I, II, and III, and the knockout is lethal after 20–90 days (McKleroy et al, 2013; Pardo et al, 2016), and MMP19 is involved in the degradation of collagen I and IV (Baidya et al, 2024; Wu et al, 2023). In addition to collagens, most collagenases can also digest other ECM and non-ECM molecules.

We report here that mice lacking the endolysosomal cation channel TRPML1 display disrupted lung function as assessed by directly measuring lung function parameters, and altered lung histology (using Sirius Red, Masson-Trichrome, and Verhoeff-Van Gieson reagents), in sum characteristic of a fibrosis-like phenotype, combined with an excess of elastin and collagen in the lung ECM. Mechanistically, defects in lysosomal exocytosis due to loss of TRPML1 and reduced extracellular levels of MMP2, MMP8, MMP9, as well as MMP12 and MMP19 in lung macrophage and fibroblast supernatants isolated from $Trpml1^{-/-}$ mice were found. These findings are strikingly complementary to results reported earlier for $Trpml3^{-/-}$ mice lacking the TRPML1-related cation channel TRPML3. In contrast to lysosomal TRPML1, TRPML3 is predominantly active in EE, and $Trpml3^{-/-}$ mice present with endocytosis defects and increased levels of MMP8 and MMP12, as well as an emphysema-like histological and lung function phenotype (with lung function parameters changed in the opposite direction as compared to $Trpml1^{-/-}$) (Spix et al, 2022). We postulate here a hitherto unknown mechanism of MMP-release dependent on TRPML1 channel activity and lysosomal exocytosis, resulting in lung fibrosis in $Trpml1^{-/-}$ mice. Our results may also have implications for Mucolipidosis type IV (MLIV) patients, who suffer from either a complete loss of TRPML1 or express TRPML1 point mutants with only residual activity or complete inactivity. Loss of functional TRPML1 results in reduced autophagy as well as reduced exocytosis, leading to the accumulation of macromolecules, including cholesterol and lactosylceramide in patient cells, but also metal ions like $Fe^{2+}$ and $Zn^{2+}$, the latter due to disrupted channel activity, in the brains of MLIV patients. MLIV patients present with a strong neurodegenerative/lysosomal storage disease (MLIV) phenotype, but are also suffering from corneal clouding and retinal degeneration as well as stomach issues (achlorhydria) and iron deficiency anemia (Bargal et al, 2001; Slaugenhaupt, 2002; Chen et al, 2014; Bach et al, 2010; Sahoo et al, 2017). MLIV patients may also be at risk of developing lung fibrosis, possibly at later stages of the disease, alongside the previously described neurodegenerative, retinal, gastric, and renal defects, including kidney fibrosis (Grieco et al, 2024).

## Results

### Loss of TRPML1 affects lung function, revealing a fibrosis-like pulmonary phenotype

We used the previously described $Trpml1^{-/-}$ mouse model, Mcoln1[tm1Sasl/J] to investigate the effects of the loss of TRPML1 on lung function (Venugopal et al, 2007) and applied the established forced oscillation technique (FlexiVent/SCIREQ) to measure lung function in mice (Spix et al, 2022; Vanoirbeek et al, 2010). $Trpml1^{-/-}$ mice showed an increase

of elastance (E), an indicator of the stiffness of the respiratory system, whereas compliance (C), the ability of the lungs to stretch, was reduced (Fig. 1A,B). Such changes of E and C are in line with a fibrosis-like phenotype and are in accordance with recent publications using the same system to characterize the bleomycin-induced fibrosis mouse model (Vanoirbeek et al, 2010). Besides elastance and compliance, other measured lung function parameters such as inspiratory capacity (IC), tissue elasticity (H), total lung capacity (A) and quasi-static compliance (Cst) were all characteristic for PF in $Trpml1^{-/-}$ mice (primewave-8 perturbation; constant-phase model) (Fig. 1B) (Spix et al, 2022; Vanoirbeek et al, 2010). The histological analysis of lung samples from WT and $Trpml1^{-/-}$ mice under basal conditions revealed enhanced Masson-Trichrome staining for collagen (green) (Fig. 1C,D), enhanced Sirius Red staining (fibrosis) (Fig. 1E,F), increased collagen antibody (Col1a1) staining (Fig. 1G,H) as well as elastin accumulation in $Trpml1^{-/-}$ samples (Figs. 1I and EV1A). Elastin was visualized using Verhoeff-Van Gieson (VVG) staining. As part of the VVG staining collagen was quantified again, corroborating the results obtained before with Masson-Trichrome staining or Col1a1 antibody (Fig. EV1A,B). Finally, we also quantified desmosine levels in BALF of WT and $Trpml1^{-/-}$ mice. Desmosine is a breakdown product of elastin and is expected to be decreased when elastin degradation is reduced. Accordingly, in $Trpml1^{-/-}$ samples levels were reduced, in line with elastin accumulation (Fig. 1J). In sum, these results were indicative of a fibrosis-like lung phenotype with an elastosis component in $Trpml1^{-/-}$ mice (Jessen et al, 2021).

Next, we tested bleomycin (Bleo) in WT and $Trpml1^{-/-}$ mice. Bleo is a drug that is being used extensively in experimental toxicology to induce fibrosis (Vanoirbeek et al, 2010). Pharmacologically, Bleo is used to treat patients with, e.g., Hodgkin's lymphoma, non-Hodgkin's lymphoma, testicular cancer, ovarian cancer, or cervical cancer, and a known side effect of the treatment with Bleo, in particular when treated with high doses, is PF (Hay et al, 1991). We performed lung function analysis as described above with PBS (control buffer) and Bleo (1.5 U/kg) treated WT and $Trpml1^{-/-}$ mice. In order to minimize unnecessary suffering of the animals due to the Bleo treatment, we limited the treatment time to 10 days. According to the literature, the timepoint of 10 days coincides with the well-established transition from the inflammatory to the fibrotic phase, when fibrogenesis is already developing. Indeed, a fibrosis-like phenotype was well detectable in WT mice after 10 days of Bleo treatment (Moeller et al, 2008; Chaudhary et al, 2006). Bleo treatment in $Trpml1^{-/-}$ mice did not result in a further exacerbation and was, after 10 days not different from PBS-treated $Trpml1^{-/-}$ mice as well as Bleo-treated WT mice, while being different from PBS-treated WT mice (Fig. 2A,B). The subsequent histological analysis of WT and $Trpml1^{-/-}$ samples after Bleo treatment revealed similar results. Sirius Red as well as Masson-Trichrome stainings pointed to a similar level of fibrosis in WT and $Trpml1^{-/-}$ mice treated with Bleo and PBS-treated $Trpml1^{-/-}$ mice (Fig. 2C–F). Again, no further exacerbation was visible in $Trpml1^{-/-}$ Bleo versus the PBS-treated group. Similar results were obtained from the stainings with a Col1a1-specific antibody (Fig. 2G,H).

Taken together, compared to WT, $Trpml1^{-/-}$ mice show changes in lung function parameters and lung histology, which are in accordance with a fibrosis-like lung phenotype. A similar phenotype was inducible in WT mice after Bleo exposure while the phenotype in $Trpml1^{-/-}$ mice resembled very much the effect obtained with the well-established fibrosis inductor Bleo.

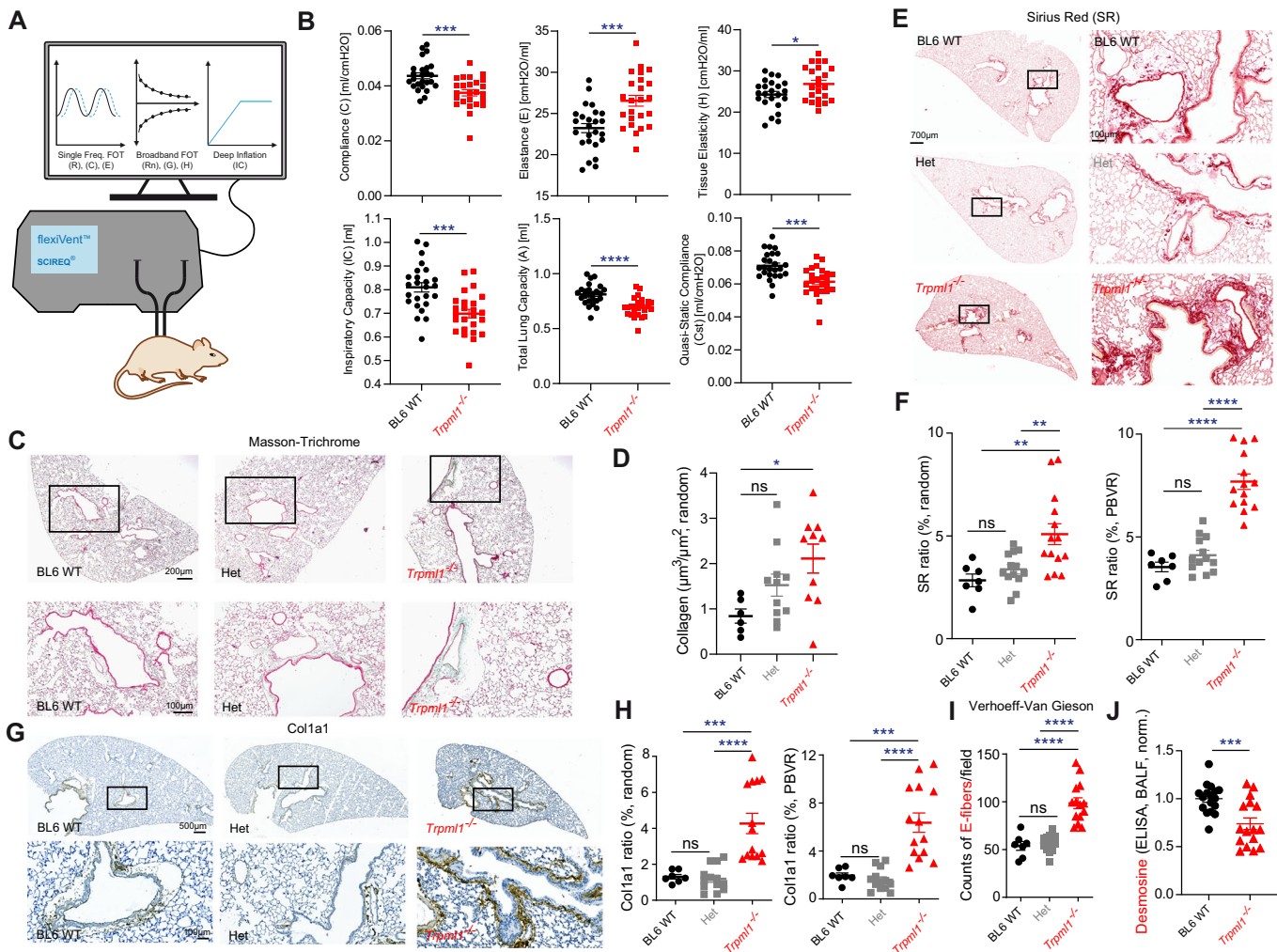

Figure 1. Lung function parameters in WT and *Trpml1⁻ᐟ⁻* mice (Mcoln1^tm1Sasl/J).

Lung function measurements were performed using the SCIREQs FlexiVent system (see Methods) (A). Different maneuvers were applied. Single frequency forced oscillation technique (FOT) allows to study the subject's response to a sinusoidal waveform, obtaining parameters such as elastance (E) and compliance (C). Broadband FOT measures the subject's response to a signal including a broad range of frequencies, below and above the subject's breathing frequency. Outcomes are, e.g., tissue elasticity (H). Deep Inflation inflates the lungs to a total capacity state. Initial and end volumes are used to calculate inspiratory capacity (IC). Pressure-volume (PV) loops capture the quasi-static mechanical properties of the respiratory system, such as quasi-static compliance (Cst) and total lung capacity (A). (B) In 3–5 months old female *Trpml1⁻ᐟ⁻* mice (*Mcoln1^tm1Sasl/J*), a significant increase of elastance (E) of the whole respiratory system was observed, whereas the compliance (C) was reduced (basal, untreated). Likewise, other lung function parameters were changed, in line with a fibrosis-like phenotype. Data were mean ± SEM. *$p < 0.05$, ***$p < 0.001$, ****$p < 0.0001$; Student's $t$-test, unpaired, two-tailed. One single dot corresponds to one mouse, each. Exact $p$ values were: Compliance, $p = 0.0003$; Elastance, $p = 0.0003$; Tissue elasticity, $p = 0.0314$; Inspiratory capacity, $p = 0.0001$; Total lung capacity, $p < 0.0001$; Quasi-static compliance, $p = 0.0002$. (C, D) Representative images, with scale bars in μm as indicated (C), and quantification of collagen deposition (D) in Masson-Trichrome-stained lung tissue sections from untreated BL6 WT and *Trpml1⁻ᐟ⁻* mouse lungs. Collagen deposition (μm³/μm²) was quantified across 30–40 randomly selected fields of view per lung (6–11 mice per group), with each point representing the mean per mouse. Data were mean ± SEM. *$p < 0.05$; One-way ANOVA followed by Tukey's post- hoc test. The exact $p$ value was $p = 0.0177$. (E–H) Representative images and quantification as mean ± SEM of Sirius Red (E, F) or Col1a1 (G, H) stained lung tissue sections from untreated BL6 WT and *Trpml1⁻ᐟ⁻* mouse lungs. Scale bars in μm as indicated. For quantification, 5–20 selected fields of view per lung (7–14 mice per group) were analysed (random, 20; peribronchovascular region (PBVR), 5–10), with each point representing the mean per mouse. **$p < 0.01$, ***$p < 0.001$, ****$p < 0.0001$; One-way ANOVA followed by Tukey's post hoc test. Data were mean ± SEM. Exact $p$ values were: SR ratio (%, PBVR), WT vs Het, $p = 0.5073$; WT vs KO, $p < 0.0001$; Het vs KO, $p < 0.0001$; SR Ratio (%, random), WT vs Het, $p = 0.6870$; WT vs KO, $p = 0.0037$; Het vs KO, $p = 0.0081$. Col1a1 ratio (%, random), WT vs Het, $p = 0.9847$; WT vs KO, $p = 0.0003$; Het vs KO, $p < 0.0001$; Col1a1 ratio (%, PBVR), WT vs Het, $p = 0.9295$; WT vs KO, $p = 0.0002$; Het vs KO, $p < 0.0001$. (I) Quantification of Verhoeff-Van Gieson-stained lung tissue sections from untreated BL6 WT and *Trpml1⁻ᐟ⁻* mouse lungs, as shown in Fig. EV1. For quantification, five selected fields of view per lung (7–14 mice per group) were analysed (counts of E-fibers per field), with each point representing the mean per mouse. ****$p < 0.0001$; One-way ANOVA followed by Tukey's post hoc test. Data were mean ± SEM. Exact $p$ values were: Counts of E-fibers, WT vs Het, $p = 0.9266$; WT vs KO, $p < 0.0001$; Het vs KO, $p < 0.0001$. (J) Quantification of the levels of desmosine in BALF isolated from WT and *Trpml1⁻ᐟ⁻* mice, using ELISA (normalized mean values ± SEM): One single dot corresponds to one biologically independent sample, i.e., one mouse. Statistical analysis was performed with Student's $t$-test, unpaired, two-tailed, ***$p < 0.001$. *Trpml1⁻ᐟ⁻* values were normalized to gender-matched WT controls within each experiment. Exact $p$ value was $p = 0.008$. The data represent the combined results of three independent experiments. Source data are available online for this figure.

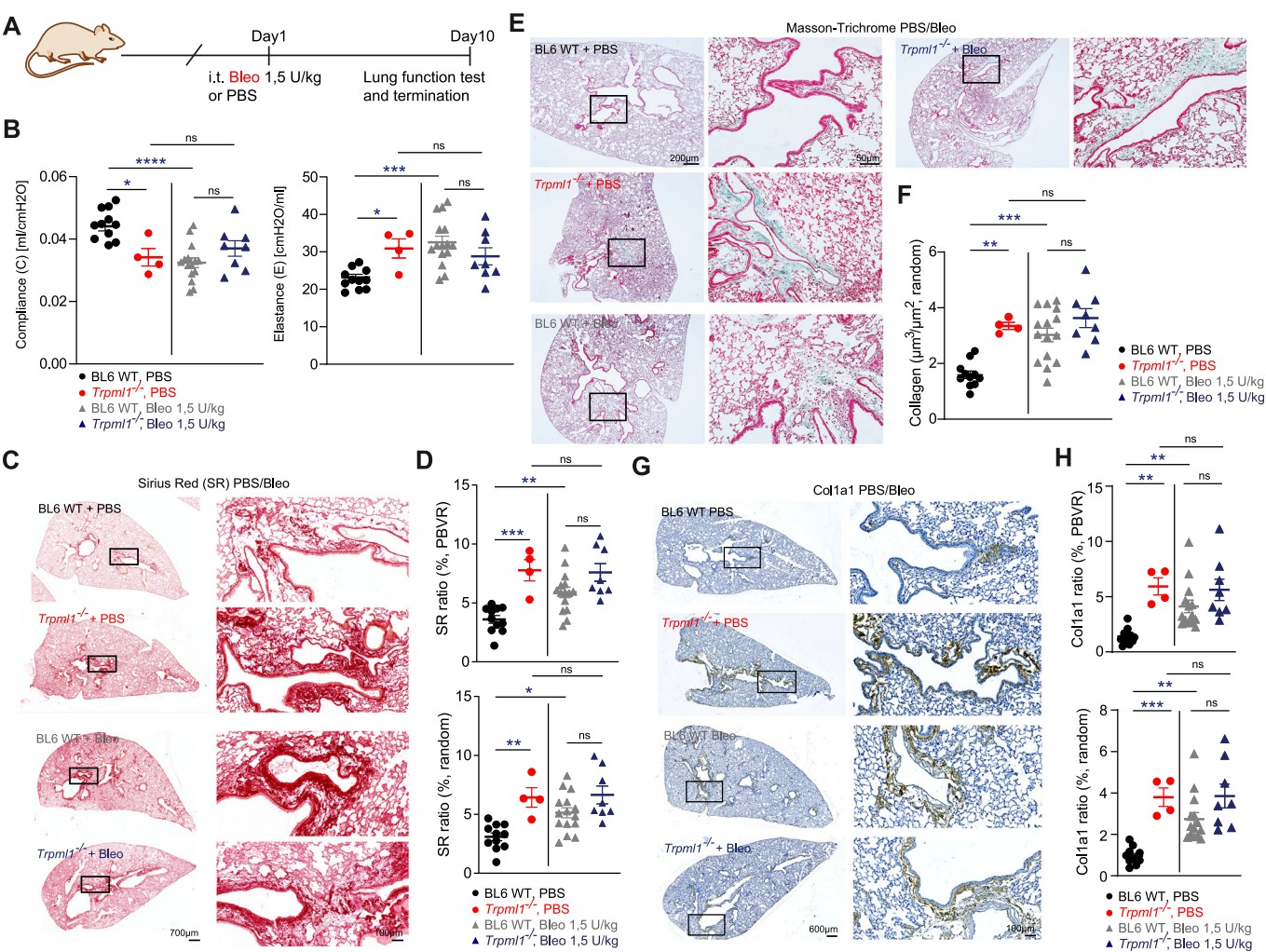

**Figure 2. Lung function parameters in WT and *Trpml1*$^{-/-}$ mice after treatment with PBS or bleomycin (Bleo).**

(A) Bleo treatment scheme. (B) Differences of E and C in PBS versus Bleo-treated (1.5 U/kg) 4–7 months old male *Trpml1*$^{-/-}$ (*Mcoln1*$^{tm1Sasl/J}$) and WT mice. Data were mean ± SEM. *$p < 0.05$, ***$p < 0.001$, ****$p < 0.0001$; One-way ANOVA followed by Holm–Šídák's post hoc test. One single dot corresponds to one mouse, each. Exact $p$ values were: Compliance, WT PBS vs KO PBS, $p = 0.0122$; WT PBS vs WT Bleo, $p < 0.0001$; Elastance, WT PBS vs KO PBS, $p = 0.0424$; WT PBS vs WT Bleo, $p = 0.0003$. (C, D) Representative images (C) and quantification as mean ± SEM (D) of Sirius Red-stained lung tissue sections from PBS or Bleo-treated BL6 WT and *Trpml1*$^{-/-}$ mouse lungs. Scale bars in μm as indicated (C). For quantification, 5–20 selected fields of view per lung (4–15 mice per group) were analysed (random, 10–20; peribronchovascular region (PBVR), 5–10), with each point representing the mean per mouse. Data were mean ± SEM. *$p < 0.05$, **$p < 0.01$, ***$p < 0.001$; One-way ANOVA followed by Holm–Šídák's post hoc test. Exact $p$ values were: SR ratio (%, PBVR), WT PBS vs KO PBS, $p = 0.0006$; WT PBS vs WT Bleo, $p = 0.0046$; SR Ratio (%, random), WT PBS vs KO PBS, $p = 0.0051$; WT PBS vs WT Bleo, $p = 0.0103$. (E, F) Representative images, with scale bars in μm as indicated (E), and quantification of collagen deposition (F) in Masson-Trichrome-stained lung tissue sections from treated versus untreated mouse lungs as shown in (B). Collagen deposition (μm³/μm²) was quantified across 30–40 randomly selected fields of view per lung (4–15 mice per group), with each point representing the mean per mouse. Data were mean ± SEM. **$p < 0.01$, ***$p < 0.001$; One ANOVA followed by Holm–Šídák's post hoc test. Exact $p$ values were: Collagen (%, random), WT PBS vs KO PBS, $p = 0.0017$; WT PBS vs WT Bleo, $p = 0.0003$. (G, H) Representative images (G) and quantification as mean ± SEM (H) of collagen Col1a1 antibody-stained lung tissue sections from PBS or Bleo-treated BL6 WT and *Trpml1*$^{-/-}$ mouse lungs. For quantification, 5–20 selected fields of view per lung (4–14 mice per group) were analysed (random, 10–20; peribronchovascular region (PBVR), 5–10), with each point representing the mean per mouse. Scale bars in μm as indicated. **$p < 0.01$, ***$p < 0.001$; One-way ANOVA followed by Holm–Šídák's post hoc test. Data were mean ± SEM. Exact $p$ values were: Col1a1 ratio (%, PBVR), WT PBS vs KO PBS, $p = 0.0016$; WT PBS vs WT Bleo, $p = 0.0055$; Col1a1 ratio (%, random), WT PBS vs KO PBS, $p = 0.0005$; WT PBS vs WT Bleo, $p = 0.0013$. Source data are available online for this figure.

## Single-cell transcriptomics, qRT-PCR, and endolysosomal patch-clamp results reveal expression of TRPML1 in lysosomes isolated from lung fibroblasts and macrophages

Four cell types in the lung are predominantly involved in PF: macrophages, fibroblasts and alveolar epithelial cells (AT1, AT2) (Selman and Pardo, 2020; Ogawa et al, 2021; Kendall and Feghali-Bostwick, 2014). The transcriptomics analysis of single-cell suspensions from whole WT mouse lungs revealed expression of TRPML1 in primary murine lung fibroblasts (pmLF) and macrophages, but not in AT1 cells, and in AT2 cells only to a limited extent in a subpopulation of cells. qRT-PCR analyses confirmed the presence of TRPML1 in macrophages and pmLF (Fig. 3A–D). In addition, we had demonstrated previously significant TRPML1

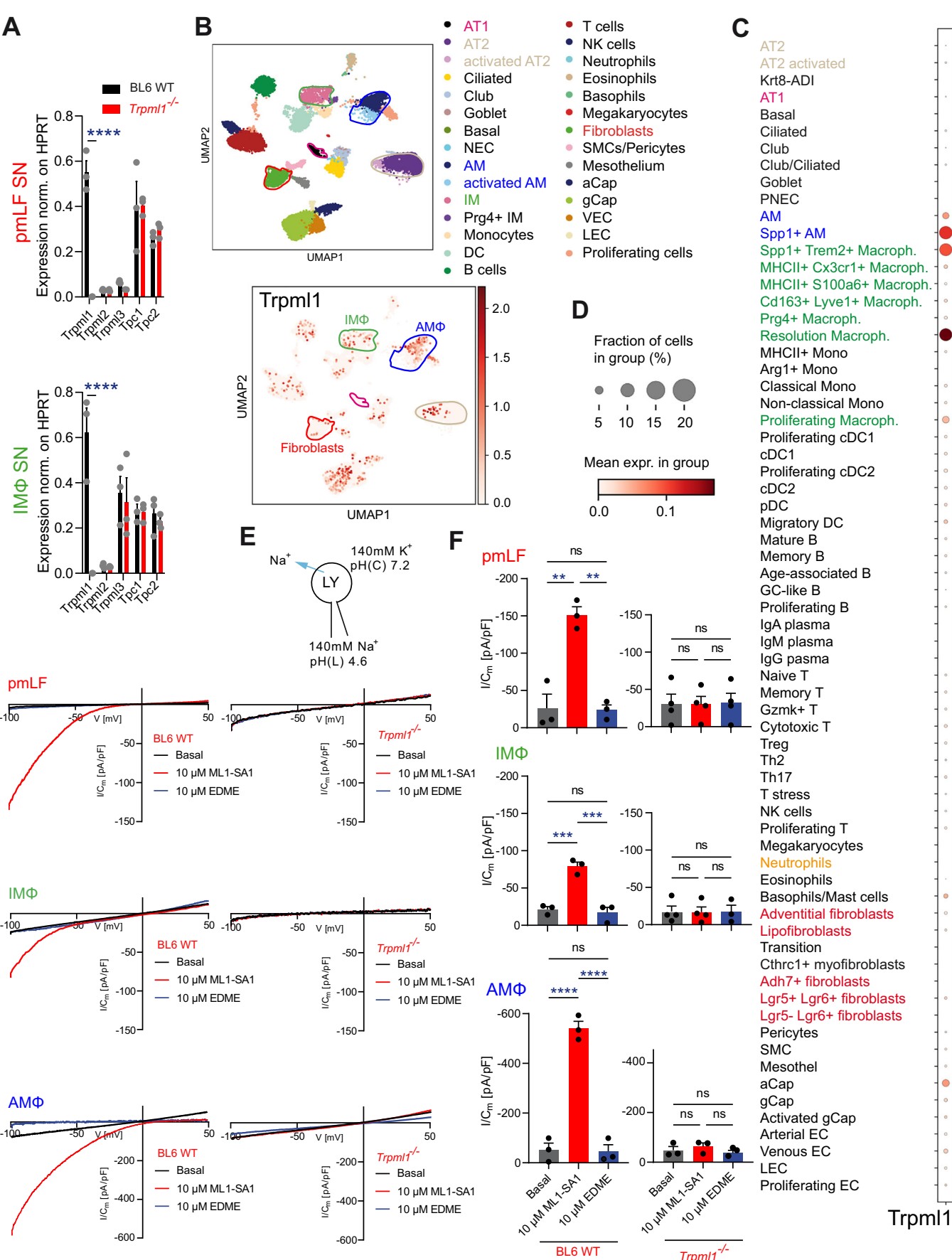

**Figure 3.  Characterization of TRPML1 expression in the lungs using single-cell transcriptomics, qRT-PCR, and endolysosomal patch-clamp.**

(A) qRT-PCR data show expression levels of *Trpml* and *Tpc* channel mRNA in pmLF and IMΦ. Data were mean ± SEM. ****$p < 0.0001$; Two-way ANOVA followed by Bonferroni's post hoc test. Each dot represents one biologically independent experiment. The exact $p$ value for the comparison of Trpml1 expression between WT and KO was $p < 0.0001$ for both pmLF and IM. (B–D) UMAP plot (B) showing clustering of single cells from control mouse lungs ($n = 9$), with cell types annotated based on transcriptional profiles. Distinct colors represent major lung cell populations, including epithelial (e.g., AT1, AT2), immune (e.g., T cells, macrophages, B cells), endothelial, stromal, and other specialized cell types. *Trpml1* expression was determined in 32 different cell types. Dotplot (C, D) shows the percentage of cells in the lung expressing *Trpml1* (coded by dot size (D)) and the average expression level of *Trpml1* based on unique molecular identifier (UMI) counts (coded by color grading). (E) Representative current density–voltage (I/C$_m$–V) traces recorded from vacuolin-enlarged late endosomes/lysosomes (LE/LY) of primary lung murine fibroblasts (pmLF) and interstitial/alveolar macrophages (IM/AMΦ), elicited by the application of 10 μM ML1-SA1 and subsequently blocked by 10 μM EDME. (F) Statistical summary of average current densities (mean ± SEM) at –100 mV from endolysosomal patch-clamp recordings as shown in (E). Each dot in the bar graph represents a single endolysosome. Statistical significance was assessed using one-way ANOVA followed by Tukey's post hoc test. **$p < 0.01$, ***$p < 0.001$, ****$p < 0.0001$. Exact $p$ values were: In pmLFs, comparing WT basal and WT 10 μM ML1-SA1, $p = 0.0012$; Comparing WT 10 μM ML1-SA1 and WT 10 μM EDME, $p = 0.0010$; In IMs, comparing WT basal and WT 10 μM ML1-SA1, $p = 0.0009$; comparing WT 10 μM ML1-SA1 and WT 10 μM EDME, $p = 0.0006$; AM, comparing WT basal and WT 10 μM ML1-SA1, $p < 0.0001$; Comparing WT 10 μM ML1-SA1 and WT 10 μM EDME, $p < 0.0001$. Source data are available online for this figure.

current activity in lysosomes isolated from alveolar macrophages (AMΦ) (Spix et al, 2022). Using the lysosomal patch-clamp technique, we now also confirmed the presence of TRPML1 in lysosomes isolated from pmLF and interstitial lung macrophages (IMΦ), alongside AMΦ (Fig. 3E,F). Furthermore, we demonstrated that other endolysosomal Ca$^{2+}$-release channels such as TRPML2, TRPML3 and the two-pore channels (TPCs) are either not expressed or not changed in their expression in *Trpml1$^{-/-}$* cells, and we confirmed the absence of TRPML1 in the knockout cells by qRT-PCR and direct current measurements (Fig. 3A).

## Screen for inflammatory mediators, TIMPs and MMPs in *Trpml1$^{-/-}$* fibroblast and macrophage supernatants and in bronchoalveolar fluid (BALF)

To elucidate the mechanism underlying the observed changes in lung function in *Trpml1$^{-/-}$* mice, we examined the levels of different inflammatory mediators as well as TIMPs and MMPs. We measured their abundance in the supernatant (SN) of pmLF and macrophages as well as in BALF using Multiplex (FirePlex) and ELISA assays, and assessed their expression levels by RNA-sequencing analysis, Western blotting, and qRT-PCR.

We started with a set of inflammatory mediators, including but not limited to a number of interleukins (IL-1β, IL-2, IL-10, IL-13, and IL-17A) as well as TNFα and MCP1/CCL2, reportedly involved in the pathophysiology of PF (She et al, 2021; Lundblad et al, 2005; Miyazaki et al, 1995; Liu et al, 2023). However, none of the 17 inflammatory mediators tested in pmLF SN and BALF (Fig. 4A) showed significant differences in WT versus *Trpml1$^{-/-}$* samples. Contrary to this, in AMΦ SN IL-17A was decreased in *Trpml1$^{-/-}$* compared to WT samples (Fig. 4A), albeit measured levels were very low. IL-17A plays a profibrotic role in the lung and as such a reduction in *Trpml1$^{-/-}$* samples is unlikely to explain the increased fibrosis observed in *Trpml1$^{-/-}$* mice (Zhang et al, 2019; Mi et al, 2011; She et al, 2021; Nie et al, 2022).

Levels of cathepsin K (CathK) and surfactant protein A (SP-A), both postulated to be involved in fibrosis development in the lung, were not different in *Trpml1$^{-/-}$* compared to WT samples (Kim et al, 2022; Bühling et al, 2004) (Fig. 4B).

TIMPs are thought to control extracellular matrix proteolysis through direct inhibition of MMPs (Arpino et al, 2015). Accordingly, increased TIMP levels would be expected to result in ECM accumulation (and fibrosis) (Leco et al, 2001). Levels of selected

TIMPs were therefore tested but found to be unchanged in the SN of cultured lung macrophages or pmLF, the latter ones expressing particularly high levels of TIMP1, 2, and 3 (Fig. 4C,D).

Finally, we tested MMPs, the natural counterparts of TIMPs, in the SN of both pmLF and lung macrophages. Among the MMPs tested in ELISA were MMP1, MMP2, MMP3, MMP7, MMP8, MMP9, MMP12, MMP13, MMP14, and MMP19. These MMPs were mainly chosen due to their demonstrated involvement in lung fibrosis and/or expression in fibrosis-relevant cell types, which at the same time express TRPML1 (i.e., pmLF and macrophages, while AT1 cells and neutrophils do not express TRPML1, and AT2 cells express TRPML1 only to a limited extent as shown in Fig. 5A,B). MMP expression analyses revealed high levels of MMP2, 3, 14, and 19 in pmLF and of MMP12 and 19 in AMΦ. MMP9, 12, and 14 showed high and MMP19 moderate expression levels in interstitial macrophages (IMΦ). MMP14 was also found to be expressed in AT1 cells (which do however not express TRPML1) and AT2 cells while MMP8 is predominantly expressed in neutrophils (which likewise do not express TRPML1), but also at lower levels in AMΦ.

Furthermore, expression of additional MMPs was analyzed, specifically, MMP10, 20, 21, 24, and 27 were found to be not or only marginally expressed in the lung (Fig. EV2), while MMP16, 17, 25, and 28 were not or only marginally expressed in pmLF, macrophages or AT1/AT2 cells. MMP11 was found to be expressed in pmLF, AT1 and AT2 cells but not in macrophages, MMP15 was found to be expressed in AT2 cells only and MMP23 in pmLF, but only in limited amounts in macrophages, AT1 and AT2 cells (Fig. EV2). Roles of the latter MMPs in lung fibrosis are, up to now, not well documented and were excluded from further analysis (Chuliá-Peris et al, 2022; Pardo et al, 2016).

Importantly, ELISA results correlated generally very well with expression data (Fig. EV2). For example, MMP7 was undetectable in AMΦ and in pmLF SNs by ELISA, in line with absent expression in the RNA-sequencing data. As another example, MMP2, 3, and 9 were undetectable in AMΦ SNs by ELISA, again concurring with the expression data. MMP1 was detectable at high levels in pmLF SN, but no differences were found between WT and *Trpml1$^{-/-}$* samples using two different assay kits (Fig. 5C). MMP3 levels in pmLF SNs were very high in both expression data and in ELISA, but also not different in *Trpml1$^{-/-}$* compared to WT (Fig. 5D). Expression of MMP13 and 14 was detectable in IMΦ, but only at low levels. Accordingly, when measuring their levels in the SN of

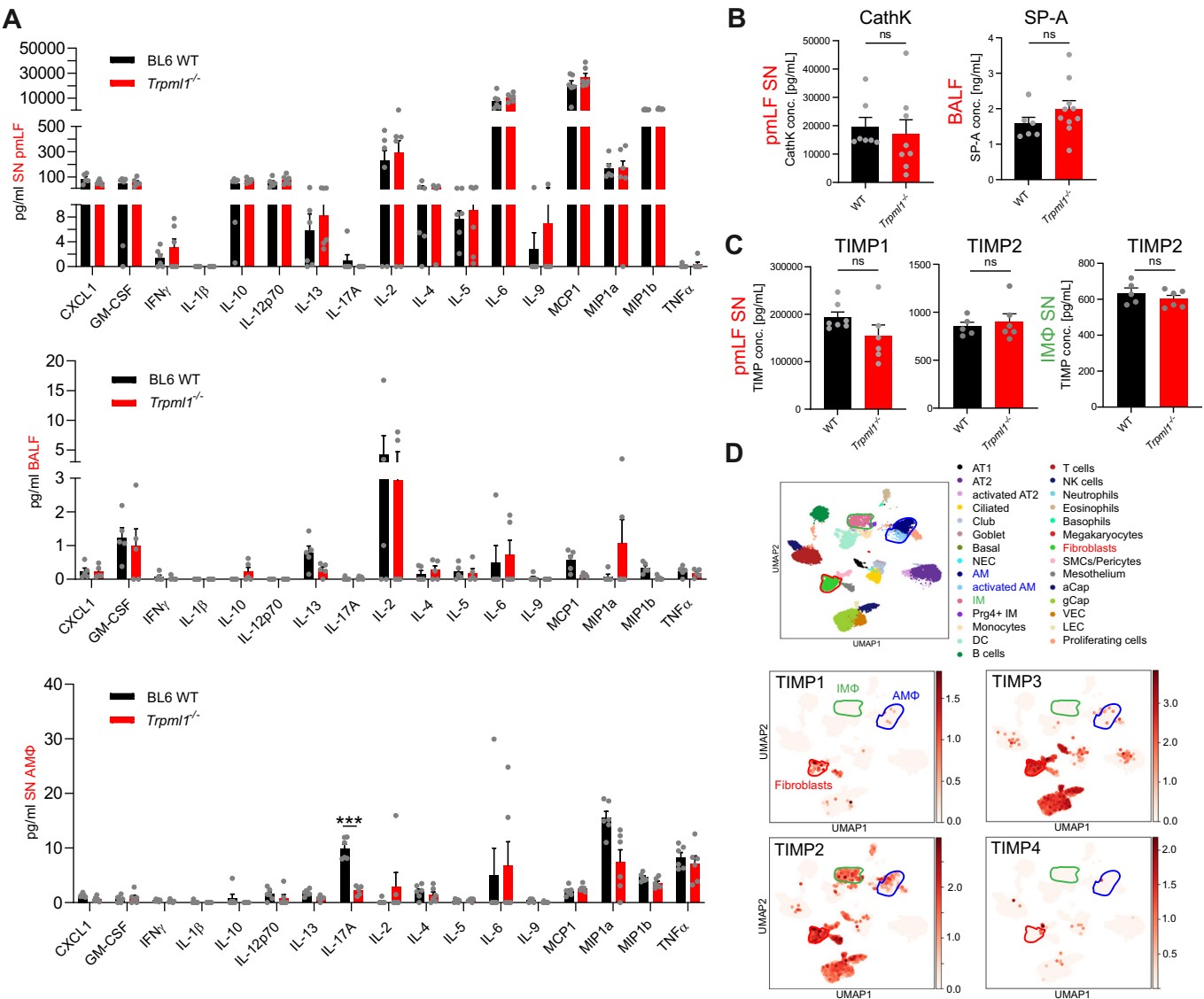

**Figure 4. Inflammatory mediator, CathK SP-A, and TIMP levels in WT and *Trpml1⁻/⁻* lung cells were measured using Multiplex and ELISA.**

(A–C) Quantification of the levels of different inflammatory mediators in BALF, macrophage, and pmLF supernatants (SN) isolated from WT and *Trpml1⁻/⁻* mice, using Multiplex (FirePlex) (A–C) as well as cathepsin K (CathK), surfactant protein A (SP-A), and TIMPs (D, E) using ELISA. BALF / pmLF SN: One single dot corresponds to one biologically independent sample i.e., one mouse. AMΦ SN: One single dot corresponds to one well. 8 WT and 8 *Trpml1⁻/⁻* mice were lavaged to obtain the appropriate number of cells for all wells. Statistical analysis of datasets in (A) was performed with multiple *t*-test, corrected for multiple comparisons using the Holm–Šídák method, ***$p < 0.001$. Data were mean ± SEM. Statistical analysis of datasets in (B, C) was performed using Student's *t*-test, unpaired, two-tailed. (D) Transcriptomics data of single-cell suspensions from whole WT mouse lungs. Featured plot shows the average expression levels of TIMPs based on unique molecular identifier (UMI) counts (coded by color grading). TIMP expression was determined in 32 different cell types. Shown are TIMP1, 2, 3, 4. Source data are available online for this figure.

WT and *Trpml1⁻/⁻* IMΦ samples using ELISA, we found that both MMP13 and MMP14 were also detectable only at low levels and levels were not different between WT and *Trpml1⁻/⁻* samples (Fig. 5E). In addition, MMP14 was detectable in pmLF SNs (here also two assay kits from different providers were used for testing), however again both assays revealed no significant differences between WT and *Trpml1⁻/⁻* (Fig. 5F). MMP14 was also detectable in AMΦ, but in line with previous results in pmLF and IMΦ no significant differences were observed between WT and *Trpml1⁻/⁻* AMΦ samples (Fig. 5G). In sum, no differences were found for MMP1, 3, 13, and 14.

By contrast, levels of MMP2, MMP8, MMP9, MMP12, and MMP19 showed significant reduction in *Trpml1⁻/⁻* cell SNs compared to control (Fig. 5H–N). Expression of MMP2 was very high in RNA-sequencing as well as qRT-PCR analyses in pmLF. Accordingly, ELISA results revealed high levels of MMP2 in WT pmLF SNs, which were reduced in *Trpml1⁻/⁻* samples (Fig. 5H). MMP8 was found to be reduced in *Trpml1⁻/⁻* AMΦ SN, but levels were generally low, in line with expression data and its predominant occurrence in neutrophils (Fig. 5I). MMP9 was likewise found to be reduced in *Trpml1⁻/⁻* in both IMΦ and pmLF (Fig. 5J,K). MMP12 was tested in AMΦ and IMΦ SNs (it was not

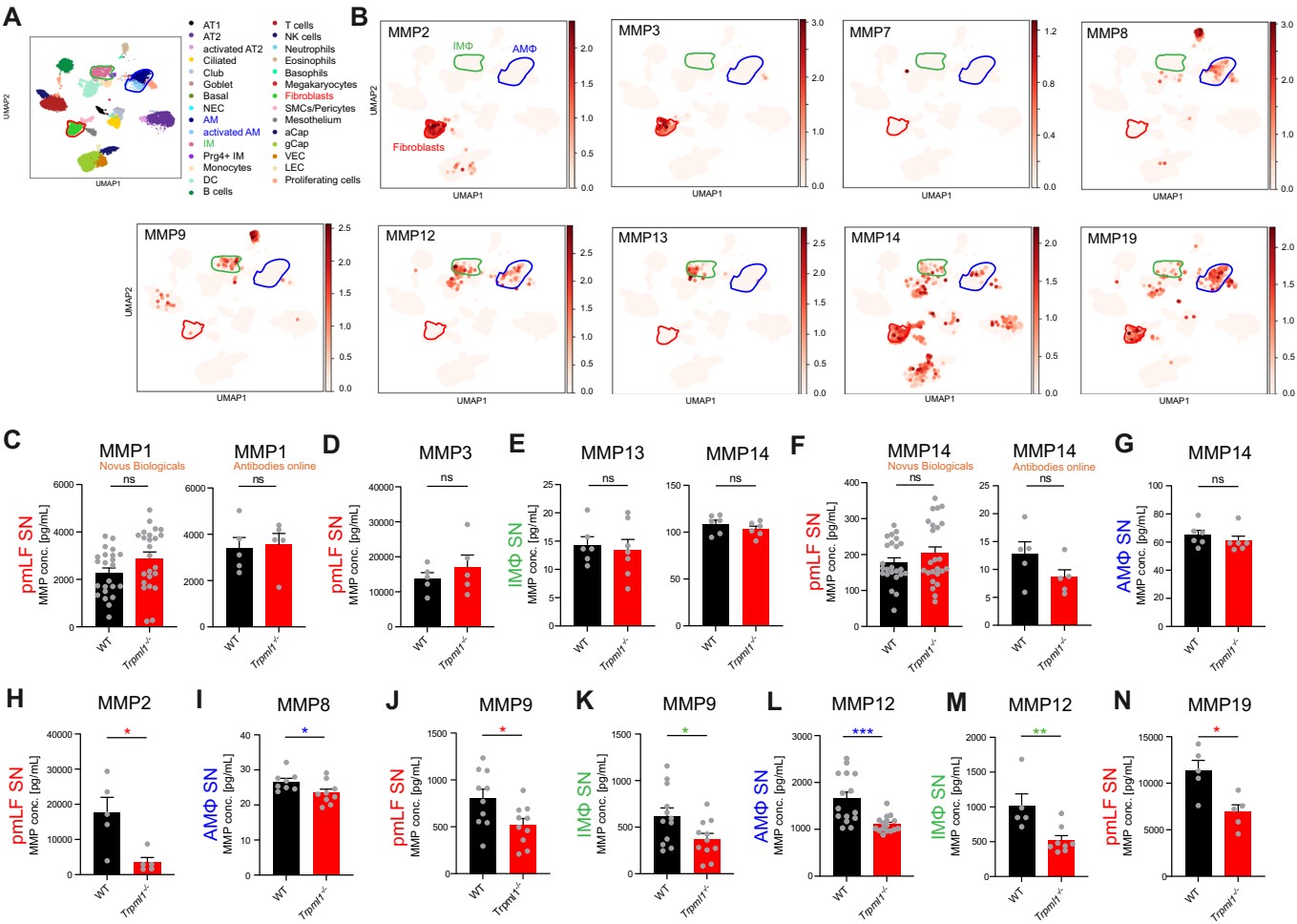

**Figure 5. MMP expression in the lung (transcriptomics) and measured MMP levels in WT and *Trpml1*⁻/⁻ lung cell SN using ELISA.**

(A, B) Transcriptomics data of single-cell suspensions from whole WT mouse lungs. The featured plot shows the average expression level of MMPs based on unique molecular identifier (UMI) counts (coded by color grading). MMP expression was determined in 32 different cell types. Shown are MMP2, 3, 7, 8, 9, 12, 13, 14, 19. (C–N) Quantification of the levels of different MMPs in macrophage (AMΦ and IMΦ) and pmLF supernatants (SN) isolated from WT and *Trpml1*⁻/⁻ mice using ELISA. IMΦ/pmLF SN: One single dot corresponds to one biologically independent sample, i.e., one mouse. AMΦ SN: One single dot corresponds to one well. Six WT and six *Trpml1*⁻/⁻ mice were lavaged to obtain the appropriate number of cells for all wells. Statistical analysis of all datasets was performed using Student's t-test, unpaired, two-tailed. *$p < 0.05$, ***$p < 0.001$. All data were mean ± SEM. Exact p values were: pmLF SN, MMP2, $p = 0.0141$; AM SN, MMP8, $p = 0.0430$; pmLF SN, MMP9, $p = 0.0241$; IM SN, MMP9, $p = 0.0327$; AM SN, MMP12, $p = 0.0002$; IM SN MMP12, $p = 0.097$; pmLF SN, MMP19, $p = 0.0124$. Source data are available online for this figure.

detectable in pmLF) and found to be reduced in *Trpml1*⁻/⁻ as well (Fig. 5L,M). Finally, expression of MMP19 was detectable with high levels in pmLF samples in ELISA, again in line with expression data, and also here a significant reduction was found in *Trpml1*⁻/⁻ samples compared to WT (Fig. 5N).

In sum, these results suggest that several MMPs are regulated independently of TRPML1 activity/expression while the levels of MMP2, MMP8, MMP9, MMP12, and MMP19 were reduced in *Trpml1*⁻/⁻ samples, hence were TRPML1 dependently regulated. Importantly, expression levels of MMPs were not changed in WT versus *Trpml1*⁻/⁻ samples (qPCR, WB), indicating that the observed reduced MMP levels were independent of changes in expression (Fig. EV2).

To evaluate the relative effects of identified MMPs on collagen degradation, we treated acutely isolated WT pmLF with siRNAs against MMP2, 9, and 19 (MMP8 was not tested as its reduction in AMΦ SN was only about 10%, while MMP12 is predominantly

relevant for elastin degradation), and assessed the effect on collagen degradation using a matrix degradation assay (zymography). We confirmed comparable knockdown levels for all three tested MMPs using qPCR, and we likewise found similar effects of the single knockdowns on collagen degradation, further corroborating that the combined reduction of several MMPs rather than a dominant effect of a single MMP causes the observed fibrosis phenotype in *Trpml1*⁻/⁻ mice (Fig. EV3A,B).

## Lysosomal exocytosis is disrupted in *Trpml1*⁻/⁻ cells

To investigate the underlying molecular mechanism leading to the extracellular reduction of MMPs, we first focused on lysosomal exocytosis, which was reported earlier to be regulated by TRPML1 (Samie et al, 2013; Di Paola and Medina, 2019; Tsunemi et al, 2019) (Fig. 6). To investigate lysosomal exocytosis, we applied the well-

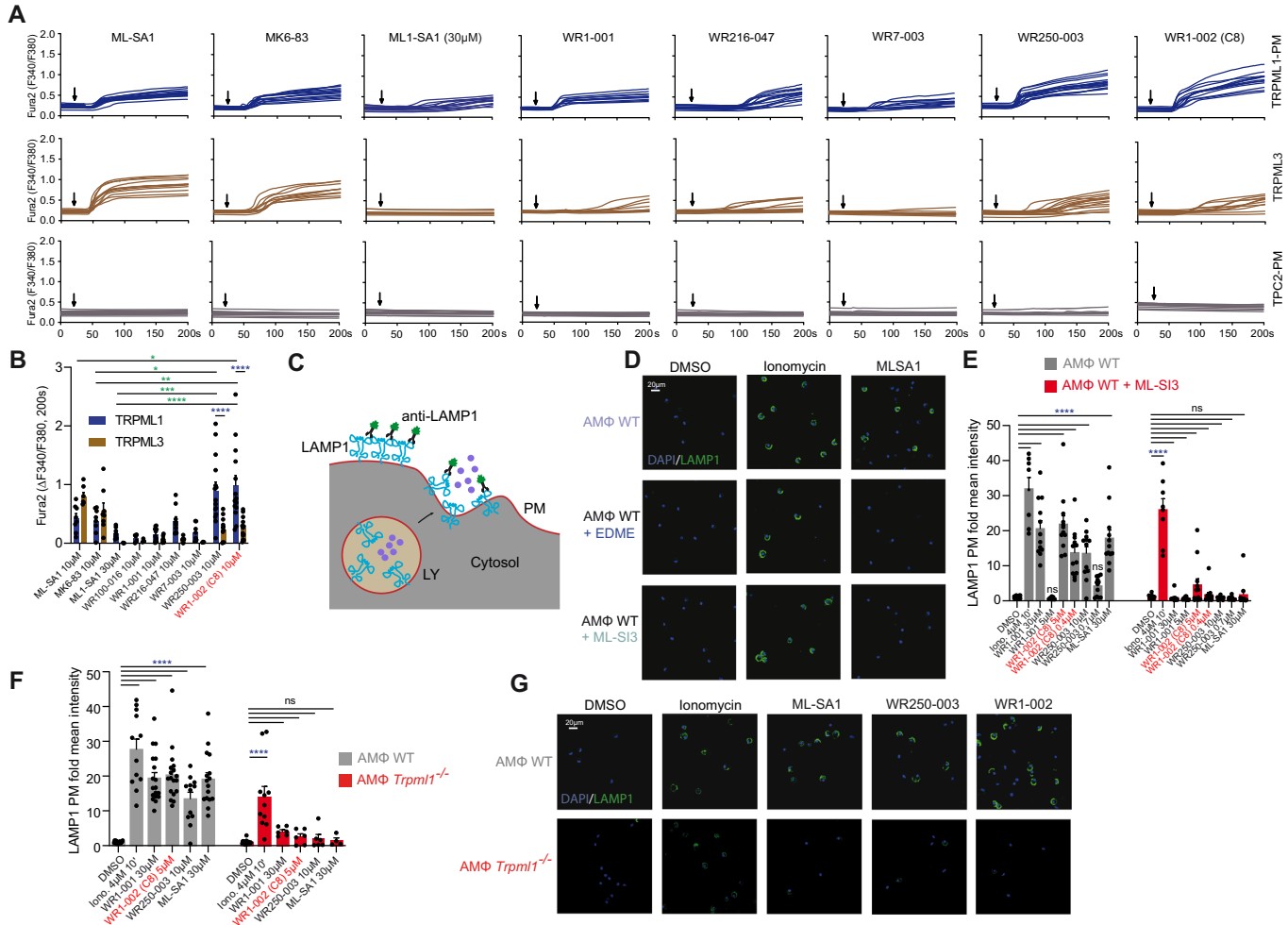

**Figure 6. Effect of different TRPML1 agonists in Ca²⁺ imaging and lysosomal exocytosis experiments.**

(A) Fura-2 Ca²⁺ imaging experiments using HEK293 cells expressing plasma membrane (PM) variants of human TRPML1 and TPC2, or TRPML3, respectively, indicating the specific levels of activation. Channels were stimulated with either ML-SA1, MK6-83, or a range of TRPML1 agonists obtained from CasmaTherapeutics, in the concentration of 10 μM, each with the exception of ML1-SA1, which was applied at a concentration of 30 μM. (B) Average values (mean ± SEM) from 7 to 10 independent experiments per condition are shown with up to ten cells, each. Statistical significance was assessed using two-way ANOVA followed by Tukey's multiple comparisons test. *$p < 0.05$, **$p < 0.01$, ***$p < 0.001$, ****$p < 0.0001$. Exact $p$ values were: Comparison of TRPML1 treated with ML-SA1 10 μM to WR1-002 10 μM, $p = 0.0130$; TRPML1 treated with MK6-83 10 μM to WR250-003 10 μM, $p = 0.0314$; TRPML1 treated with MK6-83 10 μM to WR1-002 10 μM, $p = 0.0048$; TRPML1 treated with ML1-SA1 30 μM to WR250-003 10 μM, $p = 0.0003$; TRPML1 treated with ML1-SA1 30 μM to WR1-002 10 μM, $p < 0.0001$; TRPML1 treated with WR250-003 10 μM to TRPML3 treated with WR250-003 10 μM, $p < 0.0001$; TRPML1 treated with WR1-002 10 μM to TRPML3 treated with WR1-002 10 μM, $p < 0.0001$. (C) Cartoon illustrating the LAMP1 translocation assay as used in (D–H). Upon lysosomal exocytosis, the lysosomal protein LAMP1 is detected on the plasma membrane (PM) by anti-LAMP1 antibody, followed by visualization with Alexa Fluor 488-conjugated secondary antibody. Purple dots shall represent lysosomal luminal content (e.g., MMPs) being released. (D–G) Lysosomal exocytosis experiments measuring LAMP1 translocation in WT and *Trpml1⁻/⁻* alveolar macrophages (AMΦ). Maximum effects were obtained with ionomycin (4 μM). Shown are results (mean ± SEM) obtained after 120 min treatment with DMSO, ML-SA1 as control and additional compounds as indicated. Treatment time with ionomycin (4 μM) was 10 min. Representative images of LAMP1 translocation using WT and *Trpml1⁻/⁻* AMΦ are shown in (D, G). Shown in (E, F) are normalized mean values ± SEM from at least four independent experiments, each. Statistical significance was assessed using two-way ANOVA followed by Dunnett's post hoc test. ****$p < 0.0001$. For all listed compounds in (F), WT samples showed a statistically significant difference compared to the DMSO control, with an exact $p < 0.0001$. In KO samples, all compounds were not significant except for the positive control ($p < 0.0001$). For the compounds listed in panel (E), most showed statistically significant differences compared to DMSO control in WT samples ($p < 0.0001$), except for WR1-001 5 μM and WR250-003 0.7 μM, which were not significant. In KO samples, all compounds were not significant except for the positive control, which was significant ($p < 0.0001$). Source data are available online for this figure.

established lysosomal associated membrane protein type 1 (LAMP1) plasma membrane translocation assay (Spix et al, 2022; Gerndt et al, 2020). A range of novel TRPML1 channel agonists in different doses was tested (Fig. EV3C). In Ca²⁺-imaging experiments, we first assessed the efficacy of these test compounds to release Ca²⁺ in a TRPML1-dependent manner. We counter-

screened the compounds against TPC2, another lysosomal Ca²⁺/Na⁺ permeable cation channel reported previously to stimulate lysosomal exocytosis when activated in a PI(3,5)P₂ (endogenous agonist of TPCs and TRPMLs) dependent manner (Gerndt et al, 2020) (Fig. 6A,B). We additionally counter-screened against the TRPML1-related channel TRPML3. Compounds WR250-003 and

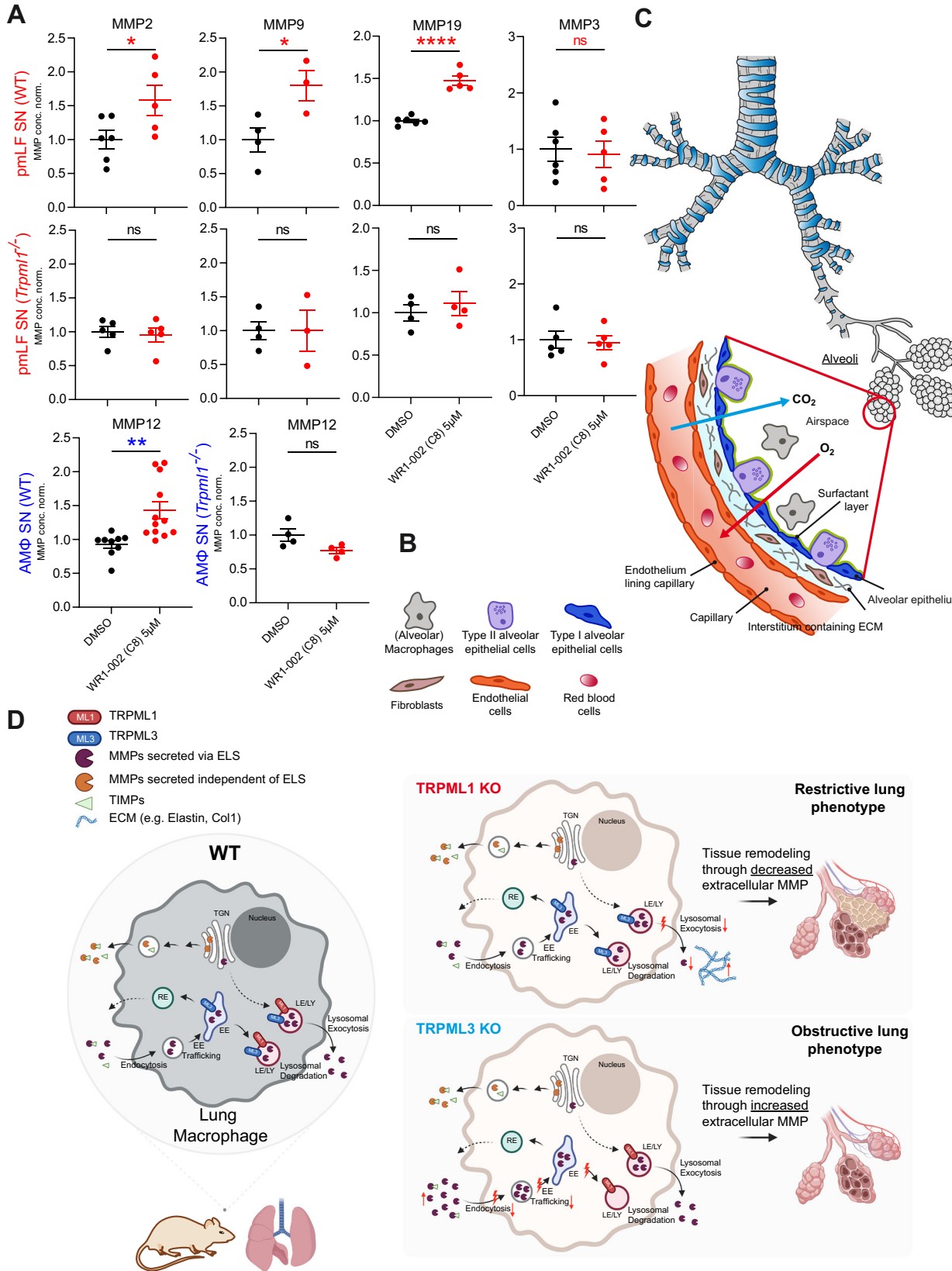

**Figure 7. Effect of TRPML1 agonist WR1-002 (C8) on MMP levels and mechanism.**

(A) Quantification of the levels of MMP2, 9, 19 in pmLF SN and MMP12 in AMΦ SN isolated from 3.5-month-old WT and *Trpml1*⁻/⁻ mice using ELISA after pretreatment with the TRPML1 agonist WR1-002 (C8). One single dot corresponds to one well. 7 WT and *Trpml1*⁻/⁻ mice were lavaged to obtain the appropriate number of cells for all wells. Statistical analysis of all datasets was performed using Student's *t*-test, unpaired, two-tailed. **$p < 0.01$. All data were mean ± SEM. Values were normalized to DMSO-treated WT controls. Exact *p* values were as follows: comparison of WT SN to WT SN treated with WR1-002 (C8) 5 μM - pmLF SN, MMP2, $p = 0.0459$; pmLF SN, MMP9, $p = 0.0371$; pmLF SN, MMP19, $p < 0.0001$; AM SN, MMP12, $p = 0.0043$. (B–D) Cartoons illustrating fibroblasts, macrophages and other cell types in the lung (B, C) as well as the trafficking and secretion dynamics of MMPs and TIMPs in WT, *Trpml1*⁻/⁻, and *Trpml3*⁻/⁻ lung macrophages (D). In WT macrophages, MMPs and TIMPs are trafficking through the trans-Golgi network (TGN) and are being secreted either directly from the TGN or via lysosomes (LY). In *Trpml1*⁻/⁻ macrophages, lysosomal exocytosis of such latter MMPs is impaired, leading to reduced MMP levels in the extracellular space. Endocytosis and intracellular trafficking remain intact, and secretion of MMPs directly from the TGN is unaffected. This disruption results in a restrictive lung phenotype. Conversely, *Trpml3*⁻/⁻ macrophages show impaired endocytosis and intracellular trafficking, but lysosomal exocytosis is preserved. This leads to the accumulation of ELS (endolysosomal system) dependent MMPs in the extracellular space, contributing to an obstructive lung phenotype. *Trpml1*⁻/⁻ and *Trpml3*⁻/⁻ show contrasting roles in regulating ELS-dependent MMP secretion and their opposing effects on lung pathology. Source data are available online for this figure.

WR1-002 ( = C8) showed the strongest effect on TRPML1 (Goodwin et al, 2021; Hooper et al, 2022). None of the compounds affected TPC2 activity (Fig. 6A). Importantly, WR250-003 and WR1-002 (C8) showed at the same time high efficacy on TRPML1 and significantly lower effects on TRPML3 as compared to other compounds. The best selectivity with no effect on either TPC2 or TRPML3 was observed with ML1-SA1, but efficacy on TRPML1 was much lower compared to WR250-003 and WR1-002 (C8) (Fig. 6B). In the next step, selected compounds were tested in the LAMP1 translocation assay using WT and *Trpml1*⁻/⁻ AMΦ or WT in combination with the TRPML1 small molecule blockers EDME (selective for TRPML1 over TRPML2/3) or ML-SI3 (Rühl et al, 2021; Leser et al, 2021). Ionomycin was used as a positive control. Lysosomal exocytosis was observed with all test compounds (Fig. 6C–H). Effects were absent or strongly reduced in the presence of TRPML1 inhibitor or in *Trpml1*⁻/⁻ cells. The effect of ionomycin was preserved in *Trpml1*⁻/⁻ cells as well as after TRPML1 inhibitor treatment (Fig. 6D–G). The effect of WR1-002 (C8) at a concentration of 0.4 μM was comparable to the effect of ML-SA1 at 30 μM and WR250-003 at 10 μM (Fig. 6E). The experiments were repeated and the results confirmed using human THP-1 macrophages (Fig. 6H). In conclusion, TRPML1 activation potently stimulates lysosomal exocytosis, providing a potential mechanism for MMP-release, with WR1-002 (C8) being the most efficacious agonist in the LAMP1 translocation assay. The latter one was thus used for further analysis.

## No defects in endocytosis and early endosomal trafficking in *Trpml1*⁻/⁻ lung cells

In addition to exocytosis, we also examined potential defects in endocytosis and endolysosomal trafficking regulation as well as autophagy due to the loss of TRPML1 (Figs. EV4 and EV5). To assess a potential role of TRPML1 in endocytosis and early endosomal trafficking in lung cells, we first probed pmLF with fluorescent transferrin (Tf). However, we found that uptake and trafficking of Tf through the early endosomal system were unchanged in *Trpml1*⁻/⁻ compared to WT (Fig. EV4A–C). We next used fluorescent dextran probes (10 and 70 kDa) to assess endocytosis (Fig. EV4D–I). With both 10 and 70 kDa dextran (the molecular weight of MMP12 is 48 kDa), we found that endocytosis rates were comparable in *Trpml1*⁻/⁻ and WT pmLF. These results are in line with previously published data e.g., by Samie et al, (2013) (Samie et al, 2013) and by Davis et al, (2020) (Davis et al, 2020),

showing that TRPML1 is involved in large particle uptake/endocytosis only (e.g., uptake of red blood cells (RBCs) or 6-μm beads). Endocytosis of 3-μm beads instead was unaffected by TRPML1. Of note, 100 kDa typically corresponds to a particle radius of 0.01 μm, suggesting that MMP, including MMP2, 8, 9, 12, and 19 uptake, would be unaffected by TRPML1, which is supported by our data.

## Impaired autophagy in *Trpml1*⁻/⁻ cells does not affect intracellular collagen levels

Loss or block of TRPML1 had also been reported earlier and was confirmed here to result in reduced autophagy (Fig. EV5A,B) (Scotto Rosato et al, 2019; Qi et al, 2024; Cunha et al, 2024). To exclude that the accumulation of extracellular collagen was not the result of reduced autophagy followed by a build-up of intracellular collagen, eventually affecting extracellular collagen levels, we performed immunocytochemistry experiments with cultured pmLF, demonstrating that intracellular levels of collagen were unchanged in WT versus *Trpml1*⁻/⁻ samples (Fig. EV5C–G).

## Small molecule activation of TRPML1 promotes MMP-release in WT but not in *Trpml1*⁻/⁻ cells

To test whether activation of TRPML1 directly affects extracellular MMP levels, we assessed the effect of the above-selected TRPML1 agonist WR1-002 (C8) on the levels of MMP2, MMP9, MMP12, and MMP19 in WT AMΦ and pmLF SNs. To confirm the specificity of the agonist, we also assessed its effects in *Trpml1*⁻/⁻ cell SNs, and we tested the effect of TRPML1 activation on MMP3, the extracellular levels of which were found to be independent of TRPML1 activity (Fig. 7A,B). In sum, we found that the levels of all tested MMPs were increased upon TRPML1 activation in the SN of WT cells compared to DMSO control, except for the negative control MMP3, indicating that direct activation of TRPML1 is capable of increasing the extracellular levels of TRPML1-dependent MMPs (opposite of what was seen in the KO).

In sum, we found that loss of TRPML1 results in a fibrosis-like lung phenotype in mice, reduced lysosomal exocytosis and reduced extracellular levels of several MMPs (MMP2, MMP8, MMP9, MMP12, and MMP19). This is surprisingly complementary to the finding that loss of TRPML3 results in reduced endocytosis, increased extracellular MMP levels, and an opposite lung function phenotype in mice, emphysema (Fig. 7D). Importantly, small

molecule activation of TRPML1 was able to increase extracellular MMPs in WT but not in *Trpml1*$^{-/-}$ cell SNs, further corroborating the specificity of TRPML1-mediated MMP release.

## Discussion

We show here the first link between an endolysosomal cation channel, TRPML1 and the development of a lung fibrosis-like phenotype. *Trpml1*$^{-/-}$ mice display decreased extracellular levels of MMP2, MMP8, MMP9, MMP12, and MMP19, but not of other tested MMPs in the SNs of acutely isolated lung macrophages and fibroblasts. Importantly, direct activation of TRPML1 with small molecule agonists was able to increase the amount of MMPs in WT pmLF and AMΦ SNs, while there was no effect in *Trpml1*$^{-/-}$ cell SNs. Furthermore, *Trpml1*$^{-/-}$ mice showed changes in lung function and histological parameters such as collagen accumulation, all in line with a fibrosis-like phenotype. The fact that extracellular levels of other MMPs expressed in the lung were unchanged in *Trpml1*$^{-/-}$ samples suggests that those MMPs are not affected by the loss of TRPML1 and may not be trafficked and/or released through the endolysosomal system, but potentially rather directly via the Golgi network or by other endolysosome- and/or TRPML1-independent pathways (Craig et al, 2015). In addition to unchanged levels of several MMPs such as MMP1, MMP3, MMP13, and MMP14, no significant changes in the levels of a plethora of inflammatory mediators, including those reported previously to be connected to lung fibrosis (e.g., different interleukins and CathK), were seen. We postulate that the combined effect of several collagenases being reduced at the same time causes the observed fibrosis phenotype rather than the reduction of a specific MMP being dominant. The function of single MMPs may also, to a considerable extent, be compensated for by other MMPs, hence the reduction of several MMPs simultaneously may be more efficacious in evoking a fibrosis-like phenotype as compensatory effects would then be more likely to be reduced. In addition to the reduction of several important collagenases, the role of MMP12 in elastin degradation may also be a critical factor in the development of the observed phenotype. Indeed, increased elastin levels have been reported previously to play a role in fibrosis and fibrosis development (Blaauboer et al, 2014; Upagupta et al, 2018; Mariani et al, 1995; Enomoto et al, 2013; Yombo et al, 2023; Hoff et al, 1999; Hansen et al, 2016). In healthy lungs, elastin contributes to proper lung function and the elasticity of lung tissue. Elastic fibers provide the elasticity needed during inhalation and ensure efficient gas exchange in the lung. While reduced elastin levels promote emphysema, excess amounts of ECM molecules such as collagen and elastin in PF are well documented to result in progressive fibrosis, scarring, and impaired lung function by reducing lung compliance, impacting ventilation, and thus compromising gas exchange (Yombo et al, 2023; Hoff et al, 1999; Ebihara et al, 2000; Starcher et al, 1978). However, while *Trpml1*$^{-/-}$ mice show a fibrosis-like phenotype, studies on fibrosis in *Mmp12*$^{-/-}$ mice seem less conclusive, showing anti-, profibrotic or even no effects in different fibrosis models (Giannandrea and Parks, 2014). Thus, on the one hand, profibrotic roles for MMP12 were found in models of PF generated by anti-Fas antibody or by bleomycin (Kang et al, 2007; Matute-Bello et al, 2007). In both studies, however, no lung function data were provided. On the

other hand, antifibrotic effects of MMP12 were observed in a mouse model involving radiation injury followed by bone marrow transfer. While a further study using bleomycin claimed no changes in fibrosis between the two genotypes (WT versus *Mmp12*$^{-/-}$) (England et al, 2011; Manoury et al, 2006), based on the lack of differences in collagen levels in the lung in qRT-PCR experiments and the lack of changes in TIMP1 and TGF-β levels in BALF. Likewise, in that latter work, unfortunately, lung function was not assessed, neither were histological data provided. Independent of the validity of the results from these different *Mmp12*$^{-/-}$ mouse studies, it should be noted that the extracellular levels of MMP12 in *Trpml1*$^{-/-}$ mice are reduced but not completely absent (which is also what we found for the other affected MMPs), hence *Mmp12*$^{-/-}$ and *Trpml1*$^{-/-}$ mice differ in their absolute MMP12 levels. Completely absent MMP12 (and the same may hold true for the other MMPs) might well have different pathophysiological consequences as compared to relative changes in MMP12 levels, based on the very delicate balance between TIMPs, MMPs and inflammatory mediators in the lung and the reported effect of MMP12 on the activity of other MMPs (Dean et al, 2008; Matsumoto et al, 1998). Thus, MMP12 was found to affect the activity of MMP2 and MMP3, with MMP2 reportedly being rather antifibrotic and MMP3 being profibrotic (Giannandrea and Parks, 2014; Afratis et al, 2018; Radbill et al, 2011). Additionally, there is evidence for MMP9 playing antifibrotic roles (Konttinen et al, 1998; Bigg *et al*, 2007; Murphy et al, 1991, 1982; Sires et al, 1995; Welgus et al, 1990; Cabrera et al, 2007), albeit this has also been discussed controversially (Wang et al, 2024). MMP8 is mostly considered profibrotic in the literature; however, evidence is likewise often controversial (Chuliá-Peris et al, 2022; Craig et al, 2015; Giannandrea and Parks, 2014). Instead, MMP19 is consistently classified as antifibrotic (Craig et al, 2015; Yu et al, 2012; Jara et al, 2015; Afratis et al, 2018). Based on these literature data and our collective results, including those obtained from the collagen degradation assay (zymography), we would consider the reduction of antifibrotic MMP2, MMP9 and MMP19 having likely similar effects regarding the observed phenotype in *Trpml1*$^{-/-}$ mice, with MMP12 potentially playing an additional important role due to its elastase function as outlined above. The fact that several MMPs with broad collagenase and elastase activities are reduced simultaneously is presumably having more impact on ECM build-up than the sole reduction or loss of a single MMP would have, raising the important question of whether TRPML1 agonists to promote the release of this specific set of MMPs, as found here, could be beneficial for fibrosis therapy. Indeed, in the past, mostly inhibitors for MMPs believed to be profibrotic were developed for fibrosis therapy. However, on the one hand it turned out that reaching specificity was difficult and on the other hand therapeutic benefit was limited (Chuliá-Peris et al, 2022; Craig et al, 2015; Meijer et al, 2010; Gaggar et al, 2011). Activating instead antifibrotic MMPs or increasing the release of antifibrotic MMPs may be a promising alternative approach, specifically the strategy to use TRPML1 agonists to stimulate the release of the here identified specific subset of MMPs seems appealing based on the fact that loss of TRPML1 results in lung fibrosis.

Taken together, our findings of increased ECM accumulation and a fibrosis-like phenotype combined with elastosis (fibroelastosis) in *Trpml1*$^{-/-}$ mice strongly suggest that the levels of a range of MMPs in the ECM are regulated by TRPML1 and that their

release depends on functional TRPML1. TRPML1 and TRPML3, as reported previously thus appear to be key factors in controlling exocytosis (TRPML1) and endocytosis (TRPML3) of several MMPs, resulting in opposite lung function phenotypes and lung histology when these channel proteins are inactivated independently (Fig. 7D).

# Methods

### Reagents and tools table

| Reagent/resource | Reference or source | Identifier or catalog number |
| --- | --- | --- |
| **Experimental models** | | |
| Mcoln1tm1Sacl mice (*M. musculus*) | Harvard University, USA | N/A |
| HEK293 (*Homo sapiens*) | DSMZ | ACC 305 |
| **Antibodies** | | |
| FN | Abcam | ab2413 |
| Col1a1 | Cell Signaling | E8F4L |
| α-SMA | Sigma | A5228 |
| LAMP1 (rat) | Santa Cruz | sc-19992 |
| LAMP1 (mouse) | Santa Cruz | sc-20011 |
| Alexa 488 goat anti-rabbit | Thermo Fisher | A11008 |
| Alexa 488 donkey anti-mouse | Thermo Fisher | A32766 |
| Alexa 647 goat anti-mouse | Thermo Fisher | A21235 |
| Alexa 488 donkey anti-rat | Thermo Fisher | A21208 |
| MMP14 | Abcam | ab51074 |
| LC3 | Sigma | L8918 |
| ß-Actin | Cell Signaling | 4970 |
| Vinculin | Cell Signaling | 4650 |
| Anti-mouse IgG, HRP-linked Antibody | Cell Signaling | 7076S |
| Anti-rabbit IgG, HRP-linked Antibody | Cell Signaling | 7074S |
| MMP2 | Proteintech | 10373-2-AP |
| MMP9 | Abcam | ab38898 |
| MMP19 | Proteintech | 14244-1-AP |
| **Oligonucleotides and other sequence-based reagents** | | |
| PCR and genotyping primers | This study | Methods |
| Accell Non-targeting Pool | Horizon Discovery Biosciences Limited | D-001910-10-20 |
| Accell GAPD Pool | Horizon Discovery Biosciences Limited | D-001930-20-20 |
| Accell Mouse Mmp2 (17390) siRNA - SMARTpool | Horizon Discovery Biosciences Limited | E-047467-00-0010 |
| Accell Mouse Mmp19 (58223) siRNA - SMARTpool | Horizon Discovery Biosciences Limited | E-048085-00-0010 |
| Accell Mouse Mmp9 (17395) siRNA - SMARTpool | Horizon Discovery Biosciences Limited | E-065579-00-0010 |
| **Chemicals, enzymes and other reagents** | | |
| CellMask™ Orange | Thermo Fisher | C10045 |
| Phalloidin CruzFluor 555 Conjugate | Santa Cruz | sc-363794 |
| Dextran, Alexa Fluor™ 568 conjugate, 10,000 MW | Thermo Fisher | D22912 |
| Dextran, Tetramethylrhodamine conjugate, 70,000 MW | Thermo Fisher | D1818 |
| Transferrin from human serum, Alexa Fluor™ 488 conjugate | Thermo Fisher | T13342 |
| Dulbecco's Modified Eagle Medium/Nutrient Mixture F12 (DMEM/F12) | Thermo Fisher | 11320-033 21041-025 |
| Dulbecco's Modified Eagle Medium (DMEM) | Thermo Fisher | 41965-039 |
| Rosewell Park Memorial Institute (RPMI) 1640 medium | Thermo Fisher | 21875-034 11835-030 |
| Hanks' Balanced Salt Solution (HBSS) | Thermo Fisher | 14025-05 |
| FluoroBrite™ DMEM | Thermo Fisher | A18967-01 |
| Dulbecco's Phosphate Buffered Saline (DPBS) | Thermo Fisher | 14190-144 |
| Fetal Bovine Serum (FBS) | Thermo Fisher | A5256701 |
| Penicillin- Streptomycin (Pen-Strep) | Thermo Fisher | 15140-122 |
| LightCycler® 480 SYBR Green I Master Mix | Roche | 04707516001 |
| PageRuler™ Prestained Protein Ladder | Thermo Fisher | 26616 |
| PageRuler™ Plus Prestained Protein Ladder | Thermo Fisher | 26619 |
| Protein Assay Dye Reagent Concentrate | Bio-Rad | 500-0006 |
| peqGREEN DNA/RNA Dye | VWR | 732-3196 |
| EasySep™ Buffer | Stemcell Technologies | 20144 |
| Bouin's solution | Sigma-Aldrich | HT10132 |
| Weigert's iron hematoxylin | Carl Roth | X907.1 and X906.1 |
| TurboFect™ | Thermo Fisher | R0531 |
| Paraformaldehyde (PFA) | Thermo Fisher | 28906 |
| 2-Mercaptoethanol | Carl Roth | 4227.1 |
| Agarose Standard | Carl Roth | 3810.3 |
| Ammonium chloride | Merck | 1.01145.1000 |
| Bovine Serum Albumin (BSA) | Sigma-Aldrich | A2153 |
| Bafilomycin A1 | Millipore | |
| DAPI | Thermo Fisher | D1306 |
| Hoechst 33342 | Thermo Fisher | 62249 |
| Dimethyl Sulfoxide (DMSO) - anhydrous | Thermo Fisher | D12345 |
| UltraPure™ 0.5 M EDTA, pH 8.0 | Thermo Fisher | 15575-038 |
| Bleomycinsulfat | Sigma-Aldrich | B5507 |
| Ethanol | Carl Roth | 5054.4 |
| Isopropanol | Merck | 1.09634.2511 |
| Methanol | Sigma-Aldrich | 34860 |

| Reagent/resource | Reference or source | Identifier or catalog number |
| --- | --- | --- |
| Acetic Acid | Carl Roth | 3738.1 |
| holo-Transferrin human | Sigma-Aldrich | T0665-50MG |
| Liquid nitrogen ($N_2$) | Linde | |
| PhosSTOP™ Phosphatase Inhibitor Cocktail Tablets | Roche | 04906837001 |
| SIGMAFAST™ Protease Inhibitor Cocktail Tablets | Sigma-Aldrich | S8830 |
| Blotting-Grade Blocker Nonfat Dry Milk | Bio-Rad | 1706404 |
| ROTIPHORESE®Gel 30 (37.5:1) | Carl Roth | 3029.1 |
| TEMED | Carl Roth | 2367.1 |
| Ponceau S | Carl Roth | 5938.2 |
| Tween 20 | Carl Roth | 9127.1 |
| Triton X-100 | Sigma-Aldrich | X-100 |
| Mowiol® 4-88 | Sigma-Aldrich | 81381 |
| Collagenase Type I | Sigma-Aldrich | C1-28 |
| Xylol | Carl Roth | 9713.4 |
| Trypsin-EDTA (0.5%) | Thermo Fisher | 15400-054 |
| Sodium deodecyl sulfate (SDS) | Carl Roth | 2326.2 |
| ML-SI3 | LMU Munich, Franz Bracher | N/A |
| ML1-SA1 | LMU Munich, Franz Bracher | N/A |
| ML-SA1 | LMU Munich, Franz Bracher | N/A |
| MK6-83 | LMU Munich, Franz Bracher | N/A |
| WR1-002 | Casma Therapeutics | N/A |
| WR250-003 | Casma Therapeutics | N/A |
| Trizma® base (TRIS base) | Sigma-Aldrich | T6066 |
| $Na_2EDTA$ | Carl Roth | 8043.1 |
| Boric acid | Carl Roth | 5935.1 |
| Glycin | Carl Roth | 0079.4 |
| Potassium phosphate monobasic ($KH_2PO_4$) | Thermo Fisher | 447670010 |
| Entellan™ | Sigma-Aldrich | 1079610100 |
| Fura-2 | MedChem Express | F1221 |
| Vacuolin | Sanat Cruz | sc-216045 |
| 5X siRNA Buffer, 100 mL | Horizon Discovery Biosciences Limited | B-002000-UB-100 |
| Accell siRNA Delivery Media | Horizon Discovery Biosciences Limited | B-005000-100 |
| Phorbol 12-myristate 13-acetate (PMA) | Carl Roth | 3029.2 |
| Ammonium persulfate (APS) | Bio-Rad | 1610700 |
| di-Sodium hydrogen phosphate dihydrate | Sigma-Aldrich | 231-448-7 |
| 3,3′-Diaminobenzidine | Sigma-Aldrich | D12384-1G |

| Reagent/resource | Reference or source | Identifier or catalog number |
| --- | --- | --- |
| Potassium methanesulfonate (K-MSA) | Sigma-Aldrich | 83000-5G-F |
| Potassium hydroxide (KOH) | Sigma-Aldrich | 757551-5 G |
| Sodium chloride (NaCl) | Sigma-Aldrich | S9888-500G |
| Calcium chloride ($CaCl_2$) | Sigma-Aldrich | C5670-100G |
| EGTA | Carl Roth | 3054.2 |
| HEPES | Carl Roth | 6763.2 |
| Sodium methanesulfonate (Na-MSA) | Sigma-Aldrich | 304506-5 G |
| Calcium methanesulfonate (Ca-MSA) | ChemCruz | Sc-486421 |
| MES | MP biomedicals | 194835 |
| Methanesulfonic acid (MSA) | Sigma-Aldrich | 471356-25 ML |
| **Software** | | |
| GraphPad Prism v10.2.1 | https://www.graphpad.com/ | |
| ImageJ | https://imagej.net/ij/index.html | |
| ZEISS LSM880, ZEN software | Zeiss | |
| Opera Phenix Plus Harmony software 5.2 | Perkin Elmer | |
| FLUOstar Omega running Reader Control software | BMG LABTECH | |
| Odyssey FC Imaging System running ImageStudio software | LI-COR | |
| PatchMaster | v2x90.4, HEKA | |
| CellProfiler | https://cellprofiler.org/ | |
| Cytoflex LX | Beckmann Coulter | |
| Leica DMi8 microscope running the LAS X software | Leica | |
| Leica AperioVersa8 microscope running Aperio ImageScope | Leica | |
| Python | https://www.python.org/ | |
| Computer-assisted stereological toolbox (CAST) software Visopharm Integrator System (VIS), Olympus BC51 microscope | Visiopharm | |
| **Other** | | |
| RevertAid First Strand cDNA Synthesis Kit | Thermo Fisher | K1622 |
| RNeasy Plus Mini Kit | Quiagen | 74104 |
| EasySep™ Mouse CD11b Positive Selection Kit II | Stemcell Technologies | 18970 A |
| Masson-Trichrome Kit | Sigma-Aldrich | 100485 |
| Verhoeff-Van Gieson Kit | VVG Elastica | Morphisto 18553 |
| Sirius Red | Sigma-Aldrich | 365548 |
| FirePlex®-96 Key Cytokines (Mouse) Immunoassay Panel | Abcam | ab235656 |
| SP-A ELISA kit | Novus biologicals | NBP2-76693 |

| Reagent/resource | Reference or source | Identifier or catalog number |
| --- | --- | --- |
| TIMP2 ELISA kit | Abcam | ab227893 |
| MMP1 ELISA kit | Novus biologicals | NBP3-06885 |
| MMP1 ELISA kit | Antibodies online | ABIN6963621 |
| MMP2 ELISA kit | Abcam | ab254516 |
| MMP3 ELISA kit | Abcam | ab100731 |
| MMP8 ELISA kit | Abcam | ab206982 |
| MMP9 ELISA kit | R&D Systems | MMPT90 |
| MMP12 ELISA kit | Abcam | ab213878 |
| MMP13 ELISA kit | Novus biologicals | NBP3-06930 |
| MMP14 ELISA kit | Novus biologicals | NBP3-06941 |
| MMP14 ELISA kit | Antibodies online | ABIN6957687 |
| MMP19 ELISA kit | Novus biologicals | NBP3-06941 |
| CathK ELISA kit | Novus biologicals | NBP3-00426 |
| TIMP1 ELISA kit | Abcam | ab196265 |
| Desmosine ELISA kit | Cusabio | CSB-E14196m |
| Collagen Degradation/Zymography Assay Kit (Fluorometric) | Abcam | ab234624 |
| µ-Slide 8 well | ibidi | 80826 |
| PhenoPlate™ 96-well plate | Revvity | 6055302 |
| PhenoPlate™ 384-well plate | Revvity | 6057302 |
| Isoflurane CP® | CP-Pharma | V7005232.00.00 |
| Introcan®-W, 20 G | B. Braun | 4254112B |
| Immobilon®-P PVDF Membrane | Merck | IPVH00005 |
| LightCycler® 480 Multiwell Plate 96 | Roche | 04 729 692 001 |
| LightCycler® 480 Sealing Foil | Roche | 04 729 757 001 |

## Animals

C57BL/6J mice of both sexes, aged (2–7 months), were used in all experiments. TRPML1 knockout mice ($Trpml1^{-/-}$ mice) were maintained on a C57BL/6J background. All animals were housed under specific pathogen–free conditions with ad libitum access to food and water, and all procedures were approved by the government (Regierung von Oberbayern, ROB 55.2.-253. Vet 02-18-06.), the University of Munich (LMU) and the German Center for Lung Research (DZL) Institutional Animal Care Guidelines.

## Lung function tests

Pulmonary function in mice was measured using a FlexiVent system (SCIREQ, Montréal, Canada). Mice were anesthetized with ketamine-xylazine, tracheostomized and connected to the Flexi-Vent system. Mice were ventilated with a tidal volume of 10 mL/kg at a frequency of 150 breaths/min in order to reach a mean lung volume similar to that of spontaneous breathing. Testing of lung mechanical properties, including dynamic compliance, elastance, tissue elasticity, inspiratory capacity, total lung capacity and quasi-static compliance was carried out by a software-generated script that took three readings per animal. Mice that died during lung function measurements were excluded from analysis. For all basal experiments, female animals (3–5 months old) were used.

## Bleomycin mouse model

Male WT versus $Trpml1^{-/-}$ mice (4–7 months old) received a single intratracheal instillation of bleomycin (1,5 U/kg; B5507 Sigma). Control mice received a comparable volume of PBS. Lung function was measured as described above using the FlexiVent system 9 or 10 days after the application (Kolb et al, 2020). Lung tissue was taken for histological analysis. Experimental conditions, including mouse strain, age, sex, treatment protocol, and housing environment, were identical between experiments.

## Lung tissue processing and immunohistochemistry

Mouse lungs (same samples were used as for lung function experiments) were fixed at a constant pressure (20 cm fluid column) by intratracheal instillation of PBS-buffered 4% paraformaldehyde (PFA). Lung lobes were embedded in paraffin and sectioned. Serial 3-µm thick sections were stained with Masson-Trichrome (100485, Sigma), Sirius Red (365548, Sigma) in saturated picric acid for visualization of fibrotic deposition, as well as Verhoeff-Van Gieson (VVG Elastica) staining, using a staining kit from Morphisto (Cat. No. 18553). Immunohistochemistry was performed with primary antibody Col1a1 (E8F4L, Cell Signaling) followed by diaminobenzidine staining (Vector Laboratories) and counterstaining with hematoxylin. Positive stained sections were scanned with Leica AperioVersa8 microscope with a 40x lens and analysed using Leica Aperio ImageScope and ImageJ software (U.S. National Institutes of Health). For quantification of collagen (VVG and Col1a1) and Sirius Red stainings, 10–20 fields of view per mouse lung were chosen (peribronchovascular region 5–10; randomly selected 15–20) at 20x magnification. Color deconvolution followed by thresholding on ImageJ was addressed to separate and analyse different colors within a multi-stained image. The area (%) of the colored structures in contrast to the white background was determined. To quantify elastin in the VVG-stained lung tissue, 5 field of view per mouse lung were chosen at a 40x magnification and the number of elastin fibers were counted in every field. Elastic fibers are stained blue-black, collagen appears red, and other tissue elements yellow.

## Quantitative morphometry

Design-based stereology was used to analyse sections using an Olympus BX51 light microscope equipped with a computer-assisted stereological toolbox (newCAST, Visiopharm) running Visiopharm Integrator System (VIS) v.6.0.0.1765 software, on Masson-Trichrome-stained lung tissue sections as previously described (Günes Günsel et al, 2022). To quantify collagen deposition, digital images of Masson's Trichrome-stained lung sections were captured and analyzed at 40x magnification after superimposing a line grid. About 30–40 randomly selected fields of view per lung were quantified. For every field, the number of grid intersections falling on collagen-positive (blue-green) regions ($P_{stained\ area}$) and the intercepts of lines with airway and vessels were recorded ($I_{intercept}$ (A + V)). L(p) is the known line length per point (9.79 µm). Collagen volume density amount was then

calculated as: Collagen (μm³/μm²) = $\sum P_{stained\ area} \times L(p)/\sum I_{intercept}$ (A + V).

## Single-cell RNA-seq. data analysis

The single-cell RNA-sequencing dataset analyzed in this study was originally generated and processed by Conlon et al, (2020) (Conlon et al, 2020) and is publicly available through the Gene Expression Omnibus under accession number GSE151674. In the original study, the Drop-seq technique was used to profile single-cell suspensions from mouse lungs exposed to either filtered air (control) or cigarette smoke for 2–6 months. For this manuscript, we focused exclusively on the control group ($n = 9$ mice). Additionally, in Fig. 3C,D control group data are shown from a bleomycin study published recently (Gote-Schniering et al, 2025). Downstream analysis was carried out using the Scanpy package in Python, following established processing procedures. Cell type annotations from the original publication were retained. Genes of interest were visualized on UMAP embeddings using sc.pl.umap, and across cell types using scanpy.pl.dotplot.

## Isolation of lung fibroblasts, lung tissue (interstitial), and alveolar macrophages from mice

For preparation of pmLF, AMΦ and IMΦ mice were deeply anesthetized with isofluorane and killed by cervical dislocation. For isolation of pmLF and IMΦ, the lungs were flushed with a transcardial perfusion with ice-cold DPBS, then the lungs were isolated, cut into small pieces and digested with Collagenase 1 (C1-28, Sigma) for 1 h, shaking at 37 °C. Afterwards, cells were passed through a 70-μm nylon mesh and the cell suspension was centrifuged, resuspended and maintained in F12/DMEM containing 10% FBS, 100 U penicillin/mL, and 100 μg streptomycin/mL. To achieve a purer pmLF culture, the cells were passaged before experimental use. IMΦ were cocultured with the pmLF in DMEM containing 10% FBS, 100 U penicillin/mL, and 100 μg streptomycin/mL to achieve a higher cell yield after separation. To separate IMΦ from the pmLF after sufficient co-culture time, the EasySep™ Mouse CD11b Positive Selection Kit II (#18970, Stemcell) was used. After separation, the IMΦ-fraction was counted and cultured for experiments in DMEM containing 10% FBS, 100 U penicillin/mL, and 100 μg streptomycin/mL (Rajan et al, 2024). For isolation of AMΦ, the diaphragm of the lung was opened through a small cut, leading to a collapse of the lungs. After removing the tissue from the neck to expose the trachea, a small cut was made between the cartilage rings to open the trachea. A cannula (Introcan-W, 20 G x 1¼, B.Braun Melsungen AG) was carefully inserted into the trachea and fixed by a suture placed around the cannulated trachea. Using 1 mL syringes, the lungs were flushed with ~0.7 mL of ice-cold DPBS at least seven times to have a high yield of cells. Each time after infusing the DPBS into the lungs, the fluid was withdrawn carefully into the syringe and collected in a tube kept on ice. Finally, the lavage was centrifuged at 1000×g, 4 °C for 10 min, and the cell pellet was cultured in RPMI containing 10% FBS, 100 U penicillin/mL, and 100 μg streptomycin/mL.

## Preparation of BALF

BALF was obtained to perform ELISA and multiplex analysis. The lungs of Trpml1⁻/⁻ or WT mice (female or male mice WT vs.

Trpml1−/−, 2–4 months old) were lavaged by instilling the lungs with $3 \times 0.5$ mL aliquots of ice-cold, sterile DPBS supplemented with protease inhibitor (PI) (Roche, #04693132001) for ELISA/ multiplex analysis. For ELISA and Multiplex (FirePlex) measurements, the harvested BALF was centrifuged (1000×g, 10 min, 4 °C) to remove cells and cell debris. The obtained supernatant was distributed into aliquots, shock-frozen and stored at −80 °C until usage.

## Preparation of supernatant from cultured cells for ELISA and Multiplex (FirePlex) assays

AMΦ, IMΦ, and pmLF were isolated from WT and Trpml1⁻/⁻ mice (female or male mice WT vs. Trpml1⁻/⁻, 2–4 months old); AMΦ were pooled together to obtain the highest possible total cell count per genotype, and cells were seeded in the respective medium in wells of a 96-well plate or 12-well plate. After 1 day, cells were washed with medium to remove non-adherent cells before refreshing the medium with 200 μl phenol-red medium, supplemented with 10% FBS, 100 U penicillin/mL, and 100 μg streptomycin/mL. Cells were then cultured for 72 h. After 72 h of incubation, SN from all wells were collected into tubes on ice. Samples were centrifuged at 1000×g for 10 min at 4 °C and shock frozen in liquid nitrogen before transferring into a −80 °C freezer until usage. For TRPML1 activation cells were exposed to WR1-002 and DMSO as control, after 48 h of incubation for the remaining 24 h. For every condition, a blank, without cells, was prepared.

## Enzyme-linked immunosorbent assay (ELISA)

MMPs, TIMPs, Cathepsin K, and SP-A levels in different sample types as SN of pmLF/AMΦ/IMΦ or BALF (from 3 to 5 months old mice) were measured by Enzyme-linked immunosorbent assay. The different ELISAs were conducted according to the manufacturer's protocol. The samples were obtained as outlined above and were analysed undiluted or, if needed, after dilution determined in preexperiments. OD absorbance at 450 nm was detected using a microplate reader (FLUOstar Omega, BMG LABTECH). The concentrations were calculated as described in the manufacturer's protocol. The following ELISAs were used for the experiments: SP-A ELISA (NBP2-76693, Novus biologicals), Cathepsin K ELISA (NBP3-00426, Novus biologicals), TIMP1 ELISA (ab196265, Abcam), TIMP2 ELISA (ab227893, Abcam), MMP1 ELISA (NBP3-06885, Novus biologicals and ABIN6963621, Antibodies online), MMP2 ELISA (ab254516, Abcam), MMP3 ELISA (ab100731), MMP8 ELISA (ab206982, Abcam), MMP9 ELISA (MMPT90, R&D systems), MMP12 ELISA (ab213878, Abcam), MMP13 ELISA (NBP3-06930, Novus biologicals), MMP14 ELISA (NBP3-06941, Novus biologicals and ABIN6957687, Antibodies online), MMP19 ELISA (NBP3-06941, Novus biologicals), IL-17A ELISA (ab199081, Abcam), DESMOSINE ELISA (CSB-E14196m, Cusabio).

## Multiplex assays

The FirePlex®-96 Key Cytokines Immunoassay Panel (ab235656, Abcam) was conducted according to the manufacturer's protocol. pmLF SN, AMΦ SN, and BALF samples were obtained (from 2 to 5 months old mice) as outlined above and were analysed undiluted

or if needed after dilution determined in preexperiments. The assays were measured using the Cytoflex LX from Beckman Coulter. Cytokine content per sample was calculated in accordance with the manufacturer's protocol using the manufacturer's software FirePlex® Analysis Workbench.

## Whole-LE/LY manual patch-clamp experiments

Manual whole-LE/LY patch-clamp recordings were performed on vacuolin-treated cells, as previously described (Chen et al, 2017; Plesch et al, 2018). Cells were incubated with compounds at 37 °C and 5% $CO_2$ prior to recordings. Vacuolin was purchased from Santa Cruz Biotechnology (sc-216045). All compounds were washed out before patch-clamp experiments. Recordings were conducted at room temperature using an EPC-10 patch-clamp amplifier (HEKA, Lambrecht, Germany) and PatchMaster acquisition software (v2x90.4, HEKA). Data were digitized at 40 kHz and filtered at 2.8 kHz. Fast and slow capacitive transients were compensated using the amplifier's internal circuitry. Patch pipettes were fire-polished to a resistance of 4–8 MΩ. Liquid junction potentials were corrected for all experiments. To apply small-molecule agonists or antagonists, the cytoplasmic solution was completely exchanged with a solution containing the respective compounds. Current amplitudes at $-100$ mV were extracted from individual voltage ramp protocols. The cytoplasmic solution contained: 140 mM K-MSA, 5 mM KOH, 4 mM NaCl, 0.39 mM $CaCl_2$, 1 mM EGTA, 10 mM HEPES (pH adjusted to 7.2 with KOH). The luminal (pipette) solution contained: 140 mM Na-MSA, 5 mM K-MSA, 2 mM Ca-MSA, 1 mM $CaCl_2$, 10 mM HEPES, and 10 mM MES (pH adjusted to 4.6 with MSA). Voltage ramps from $-100$ to $+50$ mV were applied every 5 s from a holding potential of 0 mV. All data were analyzed using PatchMaster (HEKA) and GraphPad Prism version 10.2.3.

## Genotyping and qRT-PCR

*Trpml1^{-/-}* mice were obtained from Dr. Susan Slaugenhaupt (Harvard University, Boston, USA) and were generated as described in Venugopal et al (Venugopal et al, 2007). For genotyping of *Trpml1^{-/-}* mice, the following forward and reverse primers were used: 5′-tgaggagagccaagctcatt-3′ (sense), 5′-tcatcttcctgcctccatct-3′ (antisense) and 5′-tggctggacgtaaactcctc-3′ (antisense), expected bands 400 bp (WT), 200 bp (KO); cycling conditions: annealing temperature 58 °C, 35 cycles. Total RNA was extracted from cell lines using the RNeasy Plus Mini Kit (Qiagen) according to the manufacturer's protocol. cDNA was synthesized from total RNA with RevertAid First Strand cDNA Synthesis Kit (Thermo Scientific). Real-time quantitative Reverse Transcription PCR (qRT-PCR) was performed using LightCycler 480 SYBR Green I Master Mix (Roche) and LightCycler 480 Instrument (Roche, LightCycler 480 software v1.5.1). Reactions were carried out in duplicate or triplicate under conditions according to the manufacturer's recommendations. The following forward and reverse primers were used for Mcoln1, Mcoln2, Mcoln3, Tpcn1, Tpcn2, Col1a1, Fn1, Timp1, Timp2, Timp3, Timp4, Mmp1a, Mmp1b, Mmp2, Mmp3, Mmp9, Mmp12, Mmp13, Mmp14, Mmp19, Gapdh, and Hprt: 5′-atgtggacccagccaatgatacct -3′ (sense), 5′- tgtcttcagctg-gaagtggatggt-3′ (antisense) (Mcoln1), 5′-aatttggggtcacgtcatgc-3′ (sense), 5′-agaatcgagagacgccatcg-3′ (antisense) (Mcoln2),

5′-gagttacctggtgtggctgt-3′ (sense), 5′-tgctggtagtgcttaattgtttcg-3′ (antisense) (Mcoln3), 5′-cttcgtcttccgcatgaacta-3′ (sense), 5′-cat-aagctcctccttggacatt-3′ (antisense) (Tpcn1), 5′-taaagtaccgctccatc-tacca-3′ (sense), 5′-gcagacgttcgagtaataccag-3′ (antisense) (Tpcn2), 5′-ccaagaagacatccctgaagtca-3′ (sense), 5′-tgcacgtcatcgcacaca-3′ (antisense) (Col1a1), 5′-ggtgtagcacaacttccaattacg-3′ (sense), 5′-ggagtttccgcctcgagtct-3′ (antisense) (Fn1), 5′-tcttggttccctggcgtactct -3′ (sense), 5′- gtgagtgtcactctccagtttgc -3′ (antisense) (Timp1), 5′-agccaaagcagtgagcgagaag-3′ (sense), 5′-gccgtgtagataaactcgatgtc-3′ (antisense) (Timp2), 5′-aggatgccttctgcaactccga-3′ (sense), 5′-gtgta-gaccagagtgccaaagg-3′ (antisense) (Timp3), 5′-agcaaagaccctgctga-cactc-3′ (sense), 5′-acagagggaagagtcaaatggcg-3′ (antisense) (Timp4), 5′-aggaaggcgatattgtgctctcc-3′(sense), 5′-tggctggaaagtgt-gagcaagc-3′ (antisense) (Mmp1a), 5′-gcagttgtggaagatgccatcg-3′ (sense), 5′-ccatcaaatgtgtagaggtcacc-3′ (antisense) (Mmp1b), 5′-caaggatggactcctggcacat-3′ (sense), 5′-tactcgccatcagcgttcccat-3′ (antisense) (Mmp2), 5′-ctctggaacctgagacatcacc-3′(sense), 5′-aggagtcctgagagatttgcgc-3' (antisense) (Mmp3), 5′-gctgactacgataag-gacggca-3′ (sense), 5′-tagtggtgcaggcagagtagga-3′ (antisense) (Mmp9), 5′-cacacttcccaggaatcaagcc-3′ (sense), 5′-tttggtgacacgacg-gaacagg-3′ (antisense) (Mmp12), 5′-gatgacctgtctgaggaagacc-3′ (sense), 5′-gcatttctcggagcctgtcaac-3′ (antisense) (Mmp13), 5′-ggatg-gacacagagaacttcgtg-3′ (sense), 5′-cgagaggtagttctgggttgag-3′ (anti-sense) (Mmp14), 5′-aggcactcatggctcctgtcta-3′ (sense), 5′-tgagcatctcggctctcttcctc-3′ (antisense) (Mmp19), 5′-aggtcggtgtgaacg-gatttg -3′ (sense), 5′-tgtagaccatgtagttgaggtca-3′ (antisense) (Gapdh), 5′-gctcgagatgtcatgaaggagat-3′ (sense), and 5′-aaagaactta-tagccccccttga-3′ (antisense) (Hprt). Relative expression of target gene levels was determined by normalization against Hprt levels.

## Transferrin experiments

One day after seeding cells from female or male mice (WT vs. *Trpml1^{-/-}*, 3–5 months old) in 384-well PhenoPlates (Revvity), cells were synchronized in serum-free FluoroBrite (Gibco, #A1896701) medium for 10 min at 4 °C to inhibit non-receptor-mediated endocytosis. After a dPBS wash, cells were pulsed with 20 ug/mL Tfn-Alexa 488 (T13342, Invitrogen) in serum-free medium for 20 min at 37 °C. Then, cells were washed three times with 0.1 M Glycine-dPBS to quench free Tfn-Alexa 488. After this, cells were incubated with 20 μg/mL unconjugated Transferrin (T0665, Sigma) in serum-containing medium for designated time periods ranging from 0 to 60 min to investigate recycling kinetics. After three dPBS washes, cells were fixed in 4% PFA for 10 min, washed with dPBS, stained with Hoechst 33342 (Thermo Scientific, #62249) and Phalloidin CruzFluor 633 Conjugate (Santa Cruz Biotechnologies, sc-363796), washed again three times in dPBS and immediately imaged at the Opera Phenix Plus High-Content Screening System (Revvity) at 40x magnification, using a water objective. Analysis was performed with the built-in software Harmony (Revvity), analysing the cytoplasm intensity (cell intensity excluding the nucleus). The relative decrease of fluorescence intensity over time was analysed and normalized to the 0-min timepoint.

## Dextran 10 and 70 kDa experiments

Depending on the experiment, either Dextran 10 kDa (coupled to Alexa Fluor 568, Invitrogen, #D22912) or Dextran 70 kDa (coupled

to Tetramethylrhodamine, Invitrogen, #D1818) solutions were used. One day after seeding cells from female or male mice (WT vs. $Trpml1^{-/-}$, 3–5 months old) in 384-well PhenoPlates (Revvity), cells were incubated with Cell Mask green plasma membrane stain (Invitrogen, C37608) in FluoroBrite medium for 10 min at 37 °C. After 1 dPBS wash, cells were incubated with a 50 µg/mL solution of the respective dextran in FluoroBrite medium for time periods ranging from 0 to 60 min. After removing the dextran-containing medium, cells were washed thrice with ice-cold dPBS, fixed with 4% PFA for 10 min, washed once with dPBS and stained with Hoechst 33342 (Thermo Scientific, #62249). After three dPBS washes, cells were immediately imaged at the Opera Phenix Plus High-Content Screening System (Revvity) at 40x magnification, using a water objective. Analysis was performed with the built-in software Harmony (Revvity), analysing the cytoplasm intensity (cell intensity excluding the nucleus). The relative decrease of fluorescence intensity over time was analysed and normalized to the 0 min timepoint.

## Calcium imaging

Calcium imaging experiments were performed using an inverted Leica DMi8 live-cell microscope, with image acquisition and analysis carried out using the LAS X software suite. HEK293 cells were seeded on 25 mm glass coverslips in six-well plates and transfected for 24 h with YFP-tagged plasmids encoding TRPML1-LL/AA, TRPML3-LL/AA, or TPC2 using TurboFect™ Transfection Reagent. Prior to imaging, cells were washed once with PBS and incubated for 1 h with the ratiometric calcium indicator fura-2 AM (4 µM; ABCR AB 348887) diluted in a calcium-free buffer. The cells were then carefully rinsed with calcium-free buffer containing 2 mM EGTA before transferring the coverslips to the imaging chamber. All experiments were performed in a $Ca^{2+}$-free buffer consisting of 138 mM NaCl, 6 mM KCl, 1 mM $MgCl_2$, 10 mM HEPES, 5.5 mM D-glucose monohydrate, and 2 mM EGTA (pH adjusted to 7.4 with NaOH). To maintain a calcium-free environment during stimulation, 450 µL of buffer were gently added to the imaging chamber to avoid displacing the cells, followed by 50 µL of the corresponding TRPML1 agonist solution at the 30-s mark. Calcium signals were recorded for a total duration of 200 s. Cells were alternately excited at 340 nm ($Ca^{2+}$-bound) and 380 nm ($Ca^{2+}$-free), and the fluorescence ratio was used to monitor changes in cytosolic $Ca^{2+}$ levels. Regions of interest (ROIs) and background areas were defined at both excitation wavelengths to quantify the calcium response accurately.

## Lysosomal exocytosis assay (LAMP1 translocation assay)

AMΦ and THP-1 cells were seeded on eight-well plates (Ibidi) or 96, 384-PhenoPlates (Revvity), and cultured overnight. THP-1 cells were differentiated prior for 24 h with PMA and then left resting for 72 h before starting the assay. After one wash with PBS, cells were incubated in MEM supplemented with 10 mM HEPES and treated as follows: DMSO (vehicle control, 120 min), ionomycin (4 µM, 10 min), or indicated compounds (each for 120 min). Then cells were incubated with an anti-LAMP1 antibody (sc-19992, sc-20011, Santa Cruz) in MEM supplemented with 10 mM HEPES and 5% BSA for 20 min at 4 °C. After fixation with PFA (28906, Thermo Fisher) for 20 min, cells were incubated with Alexa Fluor 488-

conjugated secondary antibody (A21208, A32766, Thermo Fisher) for 1 h in PBS containing 5% BSA. Nuclei were stained with DAPI or NucBlue. Confocal images were acquired using the Opera phenix plus system (Revvity) with a 40x water objective and analysing the results with Harmony 5.2 (Revvity).

## Immunofluorescence stainings

pmLF were seeded onto glass coverslips. After 24 h, cells were fixed with cold 4% PFA for 20 min, followed by washing with PBS and quenching with 50 mM $NH_4Cl$. Permeabilization and blocking were performed at room temperature for 1.5 h using a solution containing 2% bovine serum albumin, 5% fetal bovine serum, and 0.3% Triton X-100 in PBS. Cells were then incubated overnight at 4 °C with a primary antibody against Col1a1 (E8F4L, Cell Signaling, 1:400) diluted in blocking buffer without Triton X-100. The next day, cells were incubated with the Alexa Fluor 488-conjugated secondary antibody (A11008, 1:400, Thermo-Fisher), followed by nuclear staining with DAPI or Hoechst. Confocal images were acquired using either a Zeiss LSM880 microscope equipped with a 40x oil objective and ZEN software (v2.3 SP1) with analysis performed with CellProfiler version 4.2.5 or the Opera Phenix Plus system (Revvity) with a 40x water-immersion objective. Image analysis was performed using Harmony 5.2 software (Revvity).

## Western blotting

pmLF cell pellets were resuspended in lysis buffer (10 mM TRIS HCl, pH 8, and 0.2% SDS) supplemented with proteinases and phosphatases inhibitor (Sigma). Total cell lysis was completed by ultrasonication. Protein concentration was determined by the Bradford method. SDS-polyacrylamide gel electrophoresis (PAGE), immunoblotting, protein visualization, membrane development and protein quantification were performed as described previously (Scotto Rosato et al, 2019). The following antibodies were used: ß-Actin (4970, Cell Signaling, 1:1000), Col1a1 (E8F4L, Cell Signaling, 1:1000), LC3 (L8918, Sigma, 1:1000), MMP2 (10373-2-AP, Proteintech, 1:300), MMP9 (ab38898, abcam, 1:500), and MMP19 (14244-1-AP, Proteintech, 1:500).

## MMP knockdown experiments and zymography

To knockdown (KD) MMPs, Accell SMARTpool siRNAs targeting MMP2 (Horizon Discovery, E-047467-00-0010), MMP9 (Horizon Discovery, E-065579-00-0010) and MMP19 (Horizon Discovery, E-048085-00-0010) were used. For controls, a non-targeting siRNA Pool (Horizon Discovery, D-001910-10-20) was used as the negative control, and Accell siRNA Pool specific for GAPDH (Horizon Discovery, D-001930-20-20) was employed as a positive control to verify knockdown efficiency. We introduced self-delivering Accell siRNA to isolated lung fibroblasts according to the manufacturer´s instructions. For the knockdown experiment, 100,000 fibroblasts were seeded in 24-well plates, and after 24 h the SMARTpool Accell siRNAs were added in 2% FBS medium for 72 h. After the 72 h incubation, cells were collected for qPCR. To ensure knockdown at the protein level, the culture medium was changed back to normal medium with 10% FBS for an additional 48 h. To assess collagen degradation by MMPs in the supernatant, a

fluorometric collagenase (collagen degradation zymography) assay kit (Abcam ab234624) was used. It employs a highly quenched collagen substrate that, upon cleavage by a suitable collagenase, releases a fluorophore. In brief, following 48 h incubation in normal 10% FBS medium without phenol-red, supernatants (SN) were collected. SN were centrifuged at $1000 \times g$ at 4 °C for 20 min, transferred to a fresh prechilled tube and kept on ice. Total protein amounts were measured using a BCA protein assay kit (Thermo Scientific, A55864) and 50 µL SN were transferred into desired wells in a black 96-well plate. For the positive control, 2 µL of enzyme positive control were diluted with 18 µL of Assay Buffer 38 and 10 µL/well were used. Sample volumes and positive control were adjusted to 50 µL/well with Assay Buffer 38. Subsequently, the fluorescence was measured using a microplate reader (FLUOstar Omega, BMG LABTECH).

## Statistical analysis

All data were presented as mean ± SEM. Statistical analyses were performed using GraphPad Prism (10.2.1). Comparisons between two groups from the same mice were analyzed using paired, two-tailed Student's $t$-tests. Comparisons between independent groups were analyzed using unpaired, two-tailed Student's $t$-tests. Comparisons among three or more groups were analyzed using a one-way ANOVA, followed by Tukey's, Holm–Šídák, or Bonferroni post hoc tests, depending on the number of comparisons and the experimental design. For multiplex experiments and RT-qPCR analysis, multiple $t$-tests with Holm–Šídák correction were used to account for multiple comparisons. Experiments with two independent factors were analyzed using two-way ANOVA, followed by Sidak's, Bonferroni or Dunnett's post hoc tests, depending on whether comparisons were made to a control group or between all groups. Statistical significance was defined as: $p < 0.05$ (*), $p < 0.01$ (**), $p < 0.001$ (***), and $p < 0.0001$ (****). Exact $p$ values are reported for all comparisons. Sample sizes are listed in the figure legends. Normalization of the data was done for KD qPCR experiments, when siRNA KD of specific MMPs were normalized to their respective controls. Furthermore, normalization was done for selected ELISA analyses comparing WT or KO samples to their corresponding treated conditions and for Western blot analyses, where protein expression was first normalized to the housekeeping gene for each sample, and then expressed as fold change relative to the mean WT control value.

## Data availability

The scRNA-seq data used in this study were not generated in this manuscript and are available in the Gene Expression Omnibus database under the accession code GSE151674. All other data supporting the findings of this study are available within the Article and Supplementary Information. Source data are provided with this paper.

The source data of this paper are collected in the following database record: biostudies:S-SCDT-10_1038-S44318-026-00712-4.

## Peer review information

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

## Acknowledgements

This work was supported, in part, by funding of the German Research Foundation DFG (GRK2338 P08 to CG and MB, and P09 to TG, SFB/TRR152 P04 to CG, P06 to CW-S, P12 to MB, P16 to AD, P15 and the German Center of Lung Research (DZL) to TG. CG is further supported by DFG GR4315/6-1, DFG GR4315/7-1, and SFB1328 A21. SK is supported by the German Research Foundation DFG, the Marie-Sklodowska-Curie Program Training Network for Optimizing Adoptive T Cell Therapy of Cancer funded by the H2020 Program of the European Union (Grant 955575), the Marie-Sklodowska-Curie Training Network for tracking and controlling therapeutic immune cells in cancer (funded by the Horizon Program of The EU, Grant 101168810), Else Kröner-Fresenius-Stiftung (IOLIN to AD and SK), the European Research Council PoC Grant 101100460, the CoG Grant 101124203, and the European Research Council Starting Grant (Grant 756017).

## Author contributions

**Eva-Maria Weiden**: Conceptualization; Data curation; Software; Formal analysis; Validation; Visualization; Methodology; Writing—review and editing. **Zala Serianz**: Conceptualization; Data curation; Formal analysis; Validation; Investigation; Visualization; Methodology; Writing—review and editing. **Yvonne Klingl**: Conceptualization; Data curation; Software; Formal analysis; Supervision; Validation; Investigation; Visualization; Methodology; Project administration. **Simone Jörs**: Conceptualization; Data curation; Software; Formal analysis; Supervision; Investigation; Visualization; Methodology. **Dawid Jaślan**: Conceptualization; Data curation; Software; Formal analysis; Supervision; Validation; Investigation; Visualization; Methodology. **Marco Keller**: Resources. **Sandra Prat Castro**: Data curation; Formal analysis. **Mane Mkhitaryan**: Data curation; Formal analysis. **Aicha Jeridi**: Resources; Data curation; Formal analysis; Supervision; Investigation; Methodology; Project administration. **Daria Briukhovetska**: Conceptualization; Data curation; Formal analysis. **Barbara Spix**: Conceptualization; Data curation; Software; Formal analysis; Validation; Investigation; Visualization; Methodology. **Anna Scotto Rosato**: Data curation; Formal analysis; Supervision. **Ahmed Agami**: Data curation; Formal analysis. **Herbert B Schiller**: Resources; Supervision. **Suhasini Rajan**: Data curation; Formal analysis; Supervision. **Johann Schredelseker**: Software; Supervision. **Giorgio Fois**: Data curation; Formal analysis. **Manfred Frick**: Supervision. **Sebastian Kobold**: Resources; Supervision. **Margarethe Klein**: Supervision; Methodology. **Fabian Geisler**: Supervision. **Jorge Garcia-Fortanet**: Resources. **Leon O Murphy**: Resources. **Franz Bracher**: Supervision. **Christian Wahl-Schott**: Resources; Supervision. **Thomas Gudermann**: Resources; Supervision. **Alexander Dietrich**: Resources; Supervision. **Martin Biel**: Resources; Supervision. **Ali, Önder Yildirim**: Resources; Supervision. **Christian Grimm**: Conceptualization; Resources; Data curation; Formal analysis; Supervision; Funding acquisition; Validation; Investigation; Visualization; Methodology; Writing—original draft; Project administration; Writing—review and editing.

Source data underlying figure panels in this paper may have individual authorship assigned. Where available, figure panel/source data authorship is listed in the following database record: biostudies:S-SCDT-10_1038-S44318-026-00712-4.

## Funding

## Disclosure and competing interests statement

SK has received honoraria from Plectonic, TCR2 Inc., Miltenyi, Galapagos, Cymab, Novartis, Regeneron, BMS, and GSK. SK is an inventor of several patents in the field of immuno-oncology. SK and SE received license fees from TCR2 Inc. and Carina Biotech. SK received research support from TCR2 Inc., Tabby Therapeutics, Catalym GmbH, Plectonic GmbH, and Arcus Bioscience for work unrelated to the manuscript. The remaining authors declare no competing interests. JG-F and LOM are employees of Casma Therapeutics, 201 Brookline Ave, Boston, MA, USA, 02215.

# Expanded View Figures

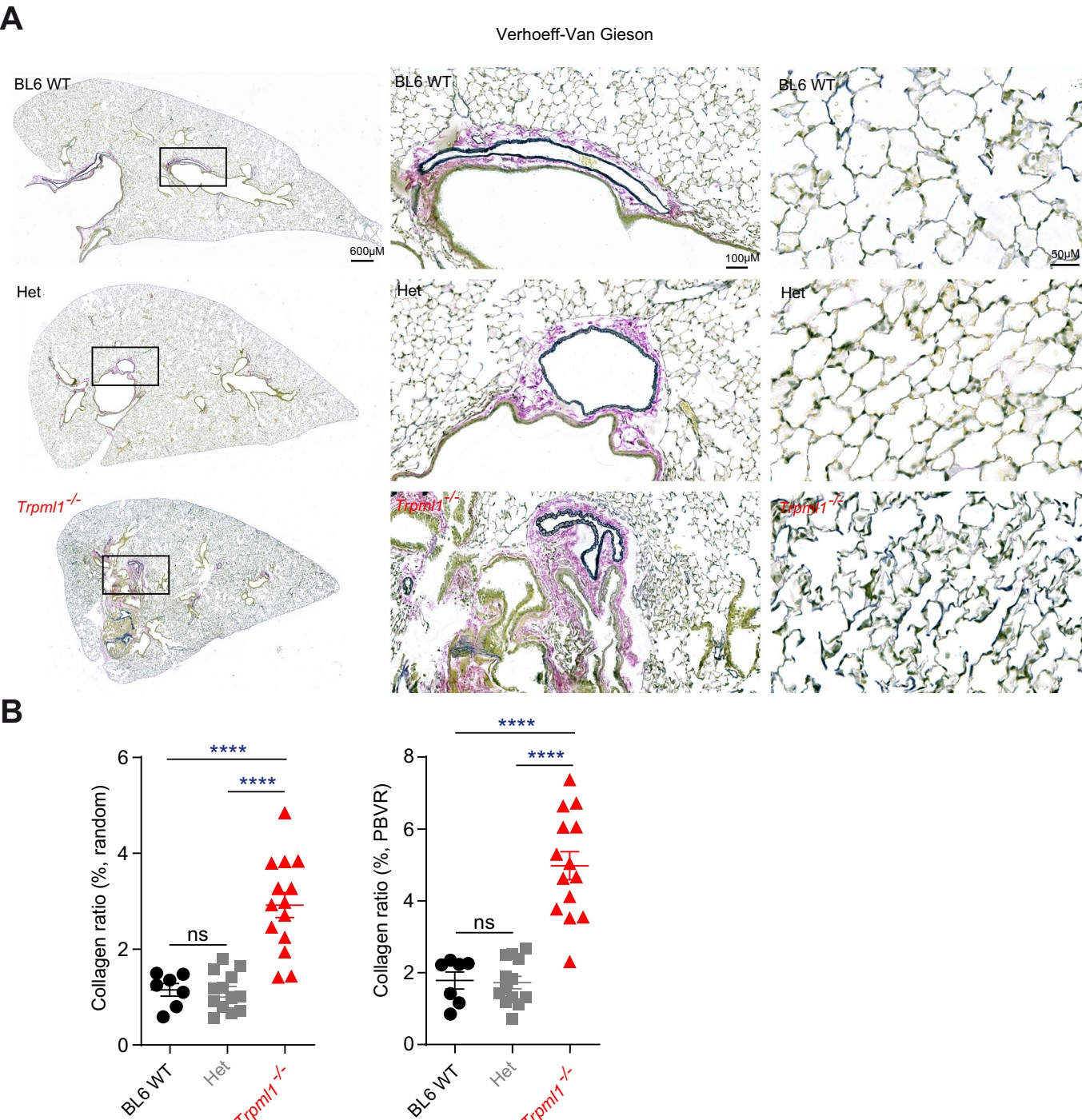

**Figure EV1.  Quantification of collagen and elastin using Verhoeff-Van Gieson staining (VVG).**

(**A**) VVG-stained lung tissue sections from untreated BL6 WT and *Trpml1*$^{-/-}$ mouse lungs. Elastic fibers are stained blue-black, collagen appears red and other tissue elements yellow. The quantification of the elastin fibers as counts per field in the VVG-stained lung tissue sections (7–14 mice per group) is shown in Fig. 1I. (**B**) Quantification of collagen as part of the VVG-stained lung tissue section analysis from untreated BL6 WT and *Trpml1*$^{-/-}$ mouse lungs as shown in (**A**). For quantification five selected fields of view per lung (7–14 mice per group) were analysed, with each point representing the mean per mouse. ***$p < 0.001$; One-way ANOVA followed by Tukey's post hoc test. Data were mean ± SEM. Exact *p* values were: Col1a1 ratio (%, random), WT vs KO, $p < 0.0001$; Het vs KO, $p < 0.0001$; Col1a1 ratio (%, PBVR), WT vs KO, $p < 0.0001$; Het vs KO, $p < 0.0001$. Source data are provided as a Source Data file. Source data are available online for this figure.

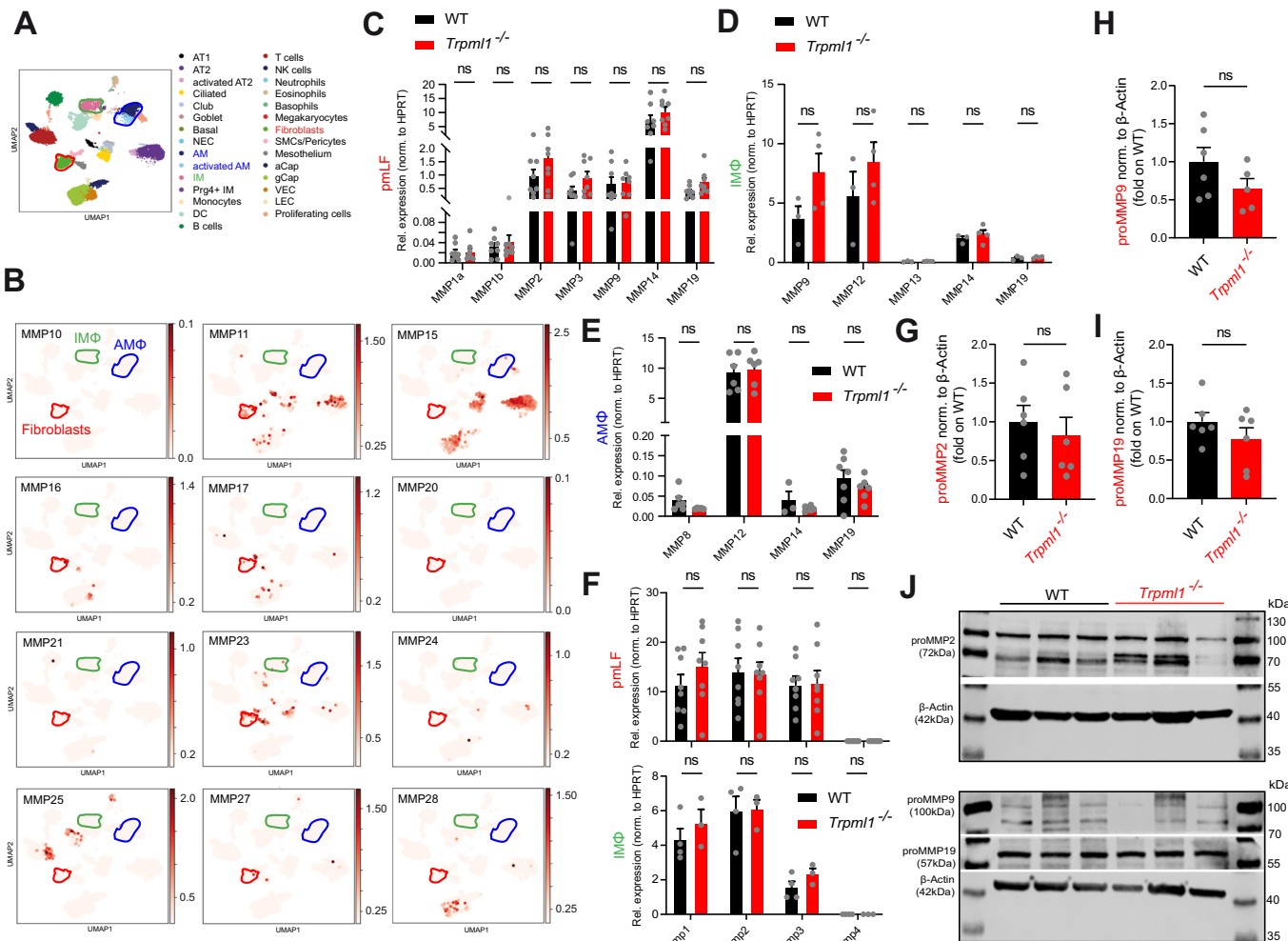

**Figure EV2. Characterization of the expression of additional MMPs in the murine lung using single-cell transcriptomics and qPCR/WB data of MMPs and TIMPs.**

(A) UMAP plot showing annotated cell clusters. Major cell types are labeled and color-coded, including epithelial, immune, endothelial, and stromal populations. MMP expression was determined in 32 different cell types. (B) UMAP plots showing the expression of selected MMP genes (MMP10, 11, 15, 16, 17, 20, 21, 23, 24, 25, 27, 28) in single cells from mouse lung tissue (filtered air group, $n = 9$). Expression is color-coded by log-normalized expression levels, with darker shades indicating higher expression. (C–E) qRT-PCR data showing mRNA expression levels of Mmp1a, Mmp1b, Mmp2, Mmp3, Mmp9, Mmp12, Mmp13, Mmp14, and Mmp19 in pmLF, IMΦ or AMΦ (WT and $Trpml1^{-/-}$). (F) qRT-PCR data showing mRNA expression levels of Timp1, Timp2, Timp3, Timp4 in pmLF and IMΦ (WT and $Trpml1^{-/-}$). In all figures, each single dot corresponds to one biologically independent sample. Data were mean ± SEM. Statistical analysis for qRT-PCR data were performed with multiple $t$-test, corrected for multiple comparisons using the Holm–Šídák method. (G–J) Western Blot analysis of different MMPs in pmLF isolated from WT and $Trpml1^{-/-}$ mice. Graphs show quantification of each MMP band normalized to ß-actin. Each single dot corresponds to cells isolated from one mouse. Data were mean ± SEM. Student's $t$-test, unpaired, two-tailed. Source data are available online for this figure.

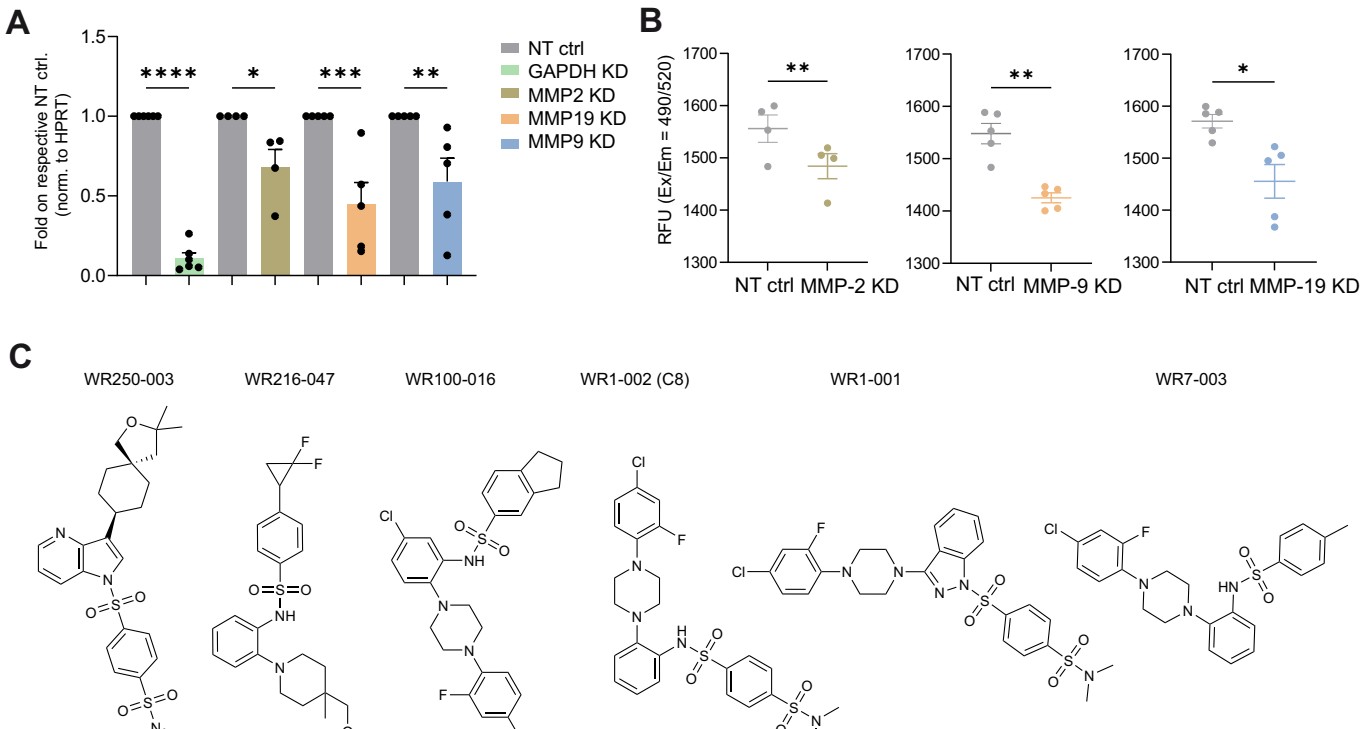

**Figure EV3. MMP knockdown experiments, zymography assay and structures of TRPML1 agonists.**

(A) Knockdown efficiency of the targeted gene determined by qPCR, shown as fold change relative to the non-targeting (NT) control and normalized to the housekeeping gene ($n = 4$–6 biological replicates). Data were mean ± SEM. Statistical significance was determined using two-way ANOVA followed by Holm–Šídák's multiple comparisons test. Shown are mean values ± SEM. **$p < 0.01$, ****$p < 0.0001$. Exact $p$ values were: GAPDH, $p < 0.0001$; MMP2, $p = 0.0154$; MMP9, $p < 0.0001$; MMP19, $p = 0.0016$. (B) Zymography-based MMP activity measured as absolute final fluorescence values, representing the total collagen degraded in each sample at reaction endpoint ($n = 4$–6 biological replicates). Data were mean ± SEM. Statistical significance was determined using a paired $t$-test, comparing non-targeting control and MMP knockdown from the same mouse. Statistical significance was defined as *$p < 0.05$ and **$p < 0.01$. Exact $p$ values were: MMP2, $p = 0.0017$; MMP9, $p = 0.0059$; MMP19, $p = 0.0413$. (C) Shown are the chemical structures of TRPML1 compounds used in this study, provided by Casma Therapeutics. Source data are available online for this figure.

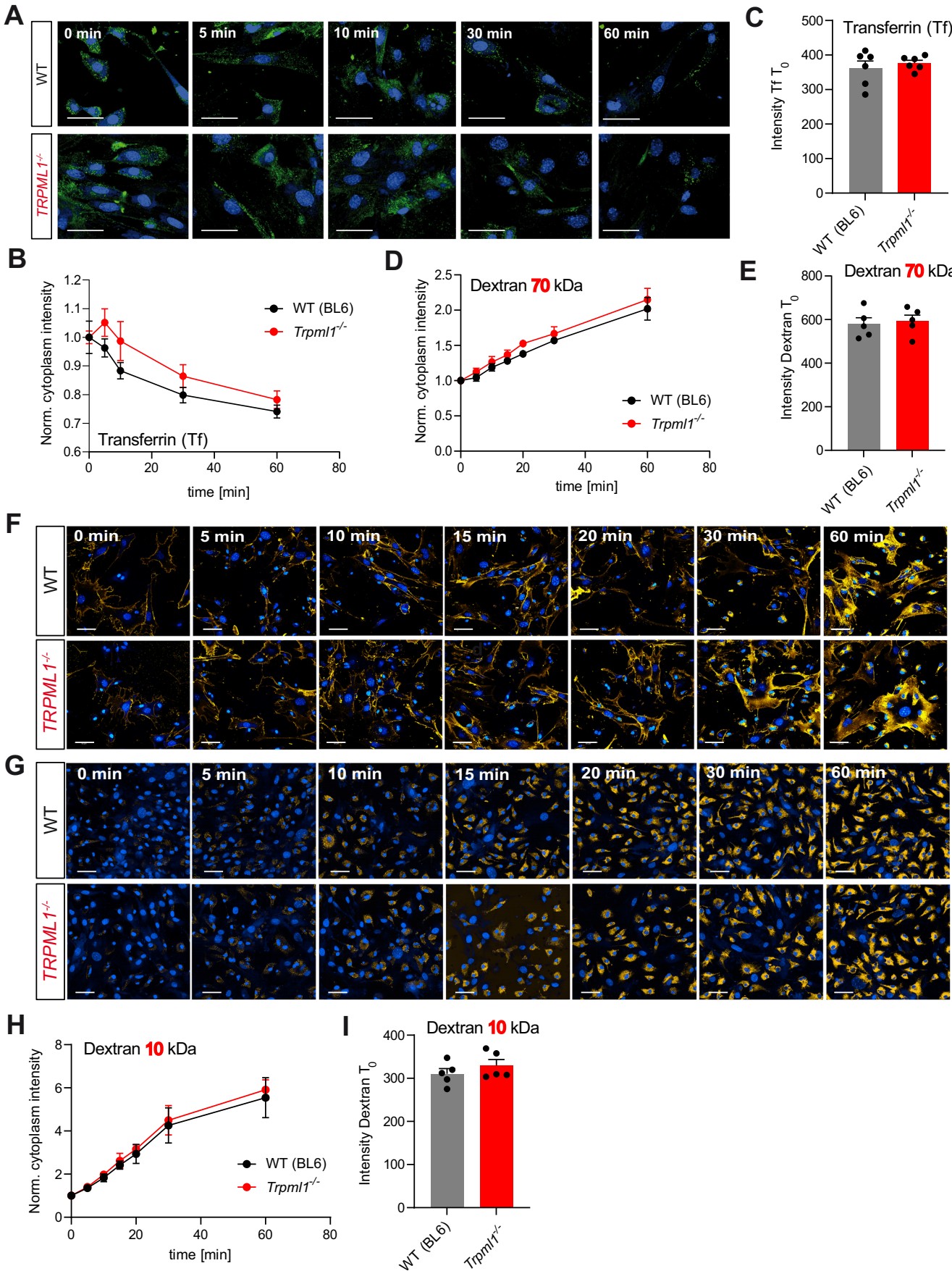

◀

**Figure EV4. Transferrin and dextran trafficking, and endocytosis in WT and *Trpml1*<sup>−/−</sup> pmLF and macrophages.**

(A–C) Transferrin (Tf) trafficking assay results. Images (scale bar 50 μm) (A) and graphs (B, C) demonstrate Tf-recycling kinetics in *Trpml1*<sup>−/−</sup> and WT pmLF, showing the decrease of Tf mean fluorescence after the 20 min pulse (measures Tf accumulation). Shown are normalized mean values ± SEM from six mice per group performed in technical triplicates. Two-way ANOVA followed by Šídák's post hoc test was applied in (B). (C) Tf mean fluorescence in pmLF (WT versus *Trpml1*<sup>−/−</sup>) after 20 min pulse with Tf Alexa Fluor 488 (0 min timepoint, measures Tf uptake). Data were shown as mean ± SEM from six mice per group performed in technical triplicates. Student's *t*-test, unpaired, two-tailed was applied. (D) Quantification of dextran 70 kDa uptake showing rates of endocytosis in *Trpml1*<sup>−/−</sup> compared to WT pmLF at various time points. Data were shown as mean ± SEM from five mice per group performed in technical triplicates. (E) Dextran 70 kDa Alexa Fluor 568 mean fluorescence in pmLF (WT versus *Trpml1*<sup>−/−</sup>) at timepoint 0 min. ($n = 4$ biological replicates). Shown are mean values ± SEM. (F–G) Representative Opera Phenix confocal images obtained from endocytosis experiments using dextran 70 and 10 kDa probes, respectively. Images show pmLF (WT versus *Trpml1*<sup>−/−</sup>) that have been treated with fluorescently labeled dextran for different time intervals. Scale bar 50 μm. (H) Quantification of dextran 10 kDa uptake showing rates of endocytosis in *Trpml1*<sup>−/−</sup> compared to WT pmLF at time points from 0 to 60 min. Two-way ANOVA followed by Šídák's post hoc test was applied. (I) Dextran 10 kDa mean fluorescence in pmLF (WT versus *Trpml1*<sup>−/−</sup>) (0 min timepoint). Data were shown as mean ± SEM from five mice per group performed in technical triplicates. Student's *t*-test, unpaired, two-tailed was applied. Source data are provided as a Source Data file. Source data are available online for this figure.

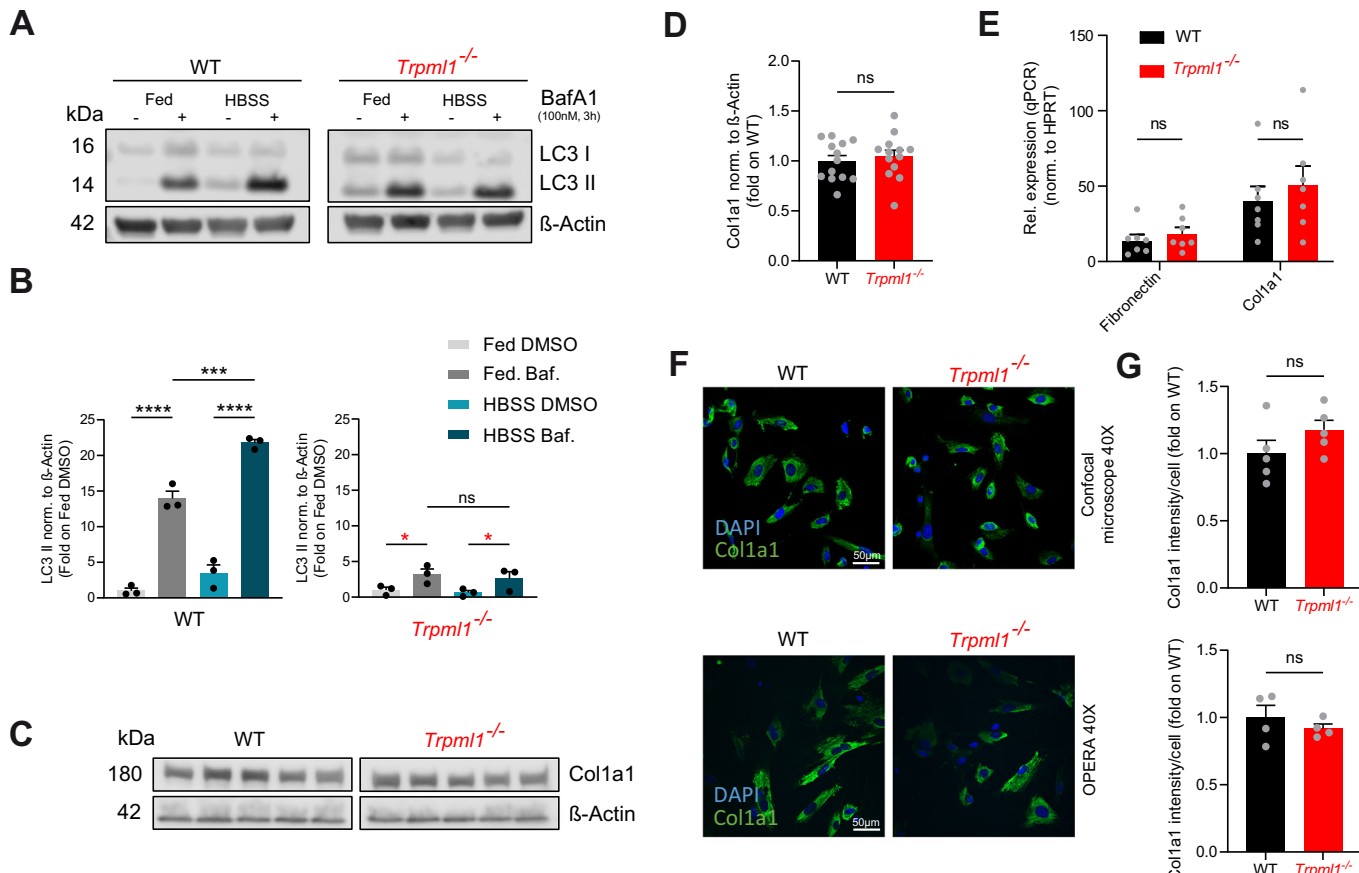

**Figure EV5. Autophagy and intracellular collagen.**

(A, B) Western Blot analysis of endogenous LC3 (LC3I-II), treated with DMSO or bafilomycin A1 (100 nM) for 3 h, under fed (complete media) or starvation (HBSS) conditions in pmLF isolated from WT and *Trpml1*$^{-/-}$ mice. Graphs show quantification of LC3-II band intensity, normalized to β-actin level. Each single dot corresponds to cells isolated from one mouse. Data were shown as mean ± SEM. *$p < 0.05$, ***$p < 0.001$, ****$p < 0.0001$; Two-way ANOVA followed by Tukey's post hoc test. Exact $p$ values were: LC3-II in WT, comparison of Fed DMSO to Fed. Baf., $p < 0.0001$; Comparison of Fed. Baf. To HBSS Baf., $p = 0.0002$; Comparison of HBSS DMSO to HBSS Baf., $p < 0.0001$; LC3-II in KO, comparison of Fed DMSO to Fed. Baf., $p = 0.0109$; Comparison of HBSS DMSO to HBSS Baf., $p = 0.0171$. (C) qRT-PCR data showing mRNA expression levels of fibronectin and Col1a1 (WT and *Trpml1*$^{-/-}$). Data were mean ± SEM. Two-way ANOVA followed by Bonferroni post hoc test. (D, E) Western blot analysis of Col1a1 in pmLF isolated from WT and *Trpml1*$^{-/-}$ mice. Graphs show quantification of Col1a1 bands normalized to ß-actin. Each single dot corresponds to cells isolated from one mouse. Data were mean ± SEM. Student's $t$-test, unpaired, two-tailed. (F, G) Confocal microscope and Opera Phenix images, and quantification of pmLF stained with Col1a1 (green) and DAPI (blue). Graphs show Col1a1 intensity per cell normalized to WT cells. Each single dot corresponds to cells isolated from one mouse. Data are mean ± SEM. Student's $t$-test, unpaired, two-tailed. Source data are provided as a Source Data file. Source data are available online for this figure.

