## [Peer Review File · The EMBO Journal]

TRPML1 suppresses pulmonary fibrosis by limiting collagen and elastin deposition

Eva-Maria Weiden, Zala Serianz, Yvonne Klingl, Simone Jörs, Dawid Jaslan, Marco Keller, Sandra Prat Castro, Mane Mkhitarian, Aicha Jeridi, Daria Briukhovetska, Barbara Spix, Anna Scotto Rosato, Ahmed Agami, Herbert Schiller, Suhasini Rajan, Johann Schredelseker, Giorgio Fois, Manfred Frick, Sebastian Kobold, Margarethe Klein, Fabian Geisler, Jorge Garcia-Fortanet, Leon Murphy, Franz Bracher, Christian Wahl-Schott, Thomas Gudermann, Alexander Dietrich, Martin Biel, Ali Yildirim, and Christian Grimm

Corresponding authors: Christian Grimm (christian.grimm@med.uni-muenchen.de) , Martin Biel (MBiel@cup.uni-muenchen.de), Ali Yildirim (oender.yildirim@helmholtz-muenchen.de)

Review Timeline:

Submission Date: Editorial	14th Jul 25
Decision:	23rd Jul 25
Appeal:	24th Jul 25
Editor's Correspondence:	30th Jul 25
Editorial Decision:	11th Sep 25
Revision Received:	26th Nov 25
Editorial Decision:	19th Jan 26
Revision Received:	20th Jan 26
Accepted:	28th Jan 26

Editor: Daniel Klimmeck

Transaction Report:

Dear Dr. Grimm,

Thank you for submitting your manuscript (EMBOJ-2025-121886) and sharing your results for consideration by The EMBO Journal. I have assessed your manuscript in detail and carefully analysed it together with my editorial colleagues. I am sorry to share that we have overall concluded that we cannot offer further pursuit by The EMBO Journal. We do encourage you to consider transfer of this manuscript to our sister venue EMBO Reports.

We appreciate your results reporting a role for calcium channel TRPML1 as a targetable gatekeeper of lung fibrosis, controlling ECM accumulation levels via regulating MMP lysosomal exocytosis. However, where the results are plausibly developed, we also noted that analogous work by your team reported relevance of control of extracellular MMP levels by related TRPML3 for lung tissue injury. Also, TRPML1 is established to regulate lysosomal exocytosis as such. Considering all information at hand, I am sorry to share that we concluded that your manuscript does not provide the sufficiently striking level of conceptual advance we have to request for publication at the EMBO Journal. We have thus regrettably decided not to send it out for peer-review.

I regret to have to disappoint you on this occasion.

Kind regards,

Daniel Klimmeck

Daniel Klimmeck, PhD
Senior Editor
The EMBO Journal

** As a service to authors, EMBO Press provides authors with the possibility to transfer a manuscript that one journal cannot offer to publish to another EMBO publication or the open access journal Life Science Alliance launched in partnership between EMBO Press, Rockefeller University Press and Cold Spring Harbor Laboratory Press. The full manuscript and if applicable, reviewers' reports, are automatically sent to the receiving journal to allow for fast handling and a prompt decision on your manuscript. For more details of this service, and to transfer your manuscript please click on Link Not Available. **

Dear Dr. Klimmeck,

thanks for your note. We were hoping that exactly because of the previous findings relating to TRPML3 our new findings that loss of TRPML1 is causing the opposite lung function pheotype (fibrosis) as compared to TRPML3, which causes emphysema and the fact that the latter is involved in endocytosis and the former in exocytosis of MMPs would make it particularly interesting as those new findings for TRML1 are both unexpected and novel, and conceptually advancing in terms of demonstrating a new mechanism of fibrosis development which depends on TRPML1? We are therefore a bit puzzled and certainly disappointed about the decision to not send it out for review. We also disagree that involvement of TRPML1 in the exocytosis of MMPs is not novel, noone has demonstrated this before and it adds important information to our overall understanding on how MMP levels are regulated in the lung.

Should your decision be irrevocable we now need to very carefully consider where to send it alternatively. It would be highly appreciated if you could send us a quick note whether based on ou clarifications above you may revoke your decision or not. Many thanks.

Kind wishes from Munich,

C. Grimm

Dear Dr. Grimm,

Thank you for your following up on our decision and detailing your rebuttal points regarding rejection of your manuscript EMBOJ-2025-121886. I have carefully evaluated these and also rediscussed the matter in the editorial team. The outcome of this consideration were that we decided to give your study the benefit of the doubt and have it formally assessed by peer-review. I will let you know about the outcome of this process shortly. Be prepared though that we will need strong support by the experts to pursue this work further post review.

Kind regards,

Daniel Klimmeck

Daniel Klimmeck, PhD
Senior Editor | The EMBO Journal
d.klimmeck@embojournal.org

Dear Dr Grimm,

Thank you again for the submission of your manuscript (EMBOJ-2025-121886) to The EMBO Journal, and providing us with a preliminary point-by-point response to the concerns raised by the referees. As mentioned, your study was assessed by two reviewers with expertise in tissue fibrosis and membrane organelles, whose comments are enclosed below.

As you will see from their comments, the referees acknowledge the analysis and potential interest and value of your findings. However, they also express major concerns, which will need to be conclusively addressed in order to proceed with publication of this study at the EMBO journal. In more detail they express substantial reservations regarding mechanistic detail provided and completeness of the data pointing to unclear cell type specificity, MMP-specific contributions, and evidence for small molecule-mediated therapeutic effects.

Given the overall interest stated and broader angle of your results, we are able to invite you to revise your manuscript experimentally to address the referees' comments, along the lines sketched in your outline. I need to stress though that we do require strong support from the referees on a revised version of the study in order to move on to publication of the work.

Please note that while well taken as such, that referee #3's point regarding examination of Trpml1 deletion at earlier time points will not be considered required for publication this study.

Please feel free to contact me if you have any questions or need further input on the referee comments.

When submitting your revised manuscript, please carefully review the instructions below.

Please feel free to approach me any time should you have additional questions related to this.

Thank you for the opportunity to consider your work for publication.

I look forward to your revision.

Kind regards,

Daniel Klimmeck

Daniel Klimmeck, PhD
Senior Editor
The EMBO Journal

Instruction for the preparation of your revised manuscript:

2) individual production quality figure files as .eps, .tif, .jpg (one file per figure).

3) a .docx formatted letter INCLUDING the reviewers' reports and your detailed point-by-point response to their comments. As part of the EMBO Press transparent editorial process, the point-by-point response is part of the Review Process File (RPF), which will be published alongside your paper.

4) a complete author checklist, which you can download from our author guidelines ([https://wol-prod-cdn.literatumonline.com/pb-assets/embo-site/Author Checklist%20-%20EMBO%20J-1561436015657.xlsx](https://wol-prod-cdn.literatumonline.com/pb-assets/embo-site/Author%20Checklist%20-%20EMBO%20J-1561436015657.xlsx)). Please insert information in the checklist that is also reflected in the manuscript. The completed author checklist will also be part of the RPF.

6) It is mandatory to include a 'Data Availability' section after the Materials and Methods. Before submitting your revision, primary datasets produced in this study need to be deposited in an appropriate public database, and the accession numbers and database listed under 'Data Availability'. Please remember to provide a reviewer password if the datasets are not yet public (see <https://www.embopress.org/page/journal/14602075/authorguide#datadeposition>).

7) Our journal encourages inclusion of *data citations in the reference list* to directly cite datasets that were re-used and obtained from public databases. Data citations in the article text are distinct from normal bibliographical citations and should directly link to the database records from which the data can be accessed. In the main text, data citations are formatted as follows: "Data ref: Smith et al, 2001" or "Data ref: NCBI Sequence Read Archive PRJNA342805, 2017". In the Reference list, data citations must be labeled with "[DATASET]". A data reference must provide the database name, accession number/identifiers and a resolvable link to the landing page from which the data can be accessed at the end of the reference. Further instructions are available at .

8) At EMBO Press we ask authors to provide source data for the main and EV figures. Our source data coordinator will contact you to discuss which figure panels we would need source data for and will also provide you with helpful tips on how to upload and organize the files.

Numerical data can be provided as individual .xls or .csv files (including a tab describing the data). For 'blots' or microscopy, uncropped images should be submitted (using a zip archive or a single pdf per main figure if multiple images need to be supplied for one panel). Additional information on source data and instruction on how to label the files are available at .

9) We replaced Supplementary Information with Expanded View (EV) Figures and Tables that are collapsible/expandable online (see examples in <https://www.embopress.org/doi/10.15252/emboj.201695874>). A maximum of 5 EV Figures can be typeset. EV Figures should be cited as 'Figure EV1, Figure EV2' etc. in the text and their respective legends should be included in the main text after the legends of regular figures.

11) For data quantification: please specify the name of the statistical test used to generate error bars and P values, the number

(n) of independent experiments (specify technical or biological replicates) underlying each data point and the test used to calculate p-values in each figure legend. The figure legends should contain a basic description of n, P and the test applied. Graphs must include a description of the bars and the error bars (s.d., s.e.m.).

We realize that it is difficult to revise to a specific deadline. In the interest of protecting the conceptual advance provided by the work, we recommend a revision within 3 months (10th Dec 2025). Please discuss the revision progress ahead of this time with the editor if you require more time to complete the revisions.

Referee #1:

The study focuses on molecular mechanisms of pulmonary fibrosis, It identifies TRPML1 as a regulator of MMP release, thus providing a novel link between lysosomal biology and ECM turnover. The findings might have potential implications for understanding pulmonary fibrosis pathogenesis and for developing TRPML1-targeted therapies. The concept is innovative and could advance the field of fibrosis research. However, further validation is necessary to firmly establish causality and therapeutic relevance.

Major Concerns:

- Introduction: The introduction would benefit from expansion to include a more comprehensive discussion of the role of ECM components and individual MMPs in lung fibrosis. Furthermore, the involvement of different lung-resident cell types in fibrosis progression should be described. A more complete summary of TRPML1 biology is also warranted to provide sufficient context for its proposed role in disease.
- Figure 1: Quantification of TRPML1 is missing. Western blotting, qPCR, or immunostaining should be included to demonstrate differences in TRPML1 expression across experimental conditions. This evidence is critical for supporting the proposed mechanistic role of TRPML1.
- Figure 2: The choice of a 10-day time point for fibrosis assessment requires justification, as fibrotic remodeling generally occurs at later stages (14-28 days). The rationale should be clearly stated, and if available, data from additional time points should be included to substantiate the conclusions.
- Cell type specificity: It remains unclear whether TRPML1-mediated regulation of MMPs is restricted to macrophages or whether other lung cell types contribute. This point should be clarified through either additional data or an expanded discussion.

Minor concerns:

- Figure set-up and formatting:
 - All figures should be presented in a uniform "letter" format.
 - Fonts should be consistent across figures, with sufficient size for readability.
 - Panel ordering should strictly follow alphabetical order in all figures.
 - In Figure 1e-f, the SR quantification should be presented after the corresponding staining images.
 - The designation *Trpml1*^{-/-} should be italicized.
 - Magnification panels should align precisely with the areas indicated in the corresponding lung panels (e.g., Figure 1).
 - Figure 2: the colors in the legends do not match the staining panels. Consistent labeling should be used.
- Figure 4: This figure shows no significant changes in inflammatory mediators and TIMPs, may be better suited for the supplementary section.
- Supplementary material: In Figure S2, a summary table of MMP expression patterns would enhance clarity and allow comparison across datasets.

Referee #2:

TRPML1 is a widely expressed Ca²⁺-permeable cation channel localized in the late endosomes and lysosomes. This channel

has been implicated in many physiological functions and diseases, most notably mucopolysaccharidosis type IV, a lysosomal storage disease caused by mutations in the TRPML1 gene. In the submitted study, by examining the effect of Trpml1 gene ablation in the mouse lung, the authors noticed an obvious pulmonary fibrosis phenotype, which was not further exacerbated by the treatment of bleomycin. This interesting observation led to further interrogation of the underlying cause of lung injury due to TRPML1 deficiency. By analyzing the expression and secretion of matrix metalloproteinases (MMPs) in TRPML1-expressing cells present in the lung, the authors concluded that TRPML1 regulates pulmonary fibrosis by promoting exocytosis of certain MMP subtypes from primary murine lung fibroblasts (pmLFs), and interstitial and alveolar macrophages (IM Φ and AM Φ). These MMPs then degrade extracellular matrix (ECM) to counter the fibrogenic process. These findings are very interesting, as they reveal a novel aspect of TRPML1 function in fibrogenesis, with potential mechanistic insights. The analysis of the pulmonary fibrosis phenotypes was thorough; the single-cell RNA-seq analysis was informative; the extensive evaluation of large numbers of inflammatory cytokines, MMPs, and TIMPs in the tissue and supernatants of cultured cells was commendable. However, despite the large amount of work, the mechanistic insights revealed based on the data shown in the manuscript are rather incomplete. The major issues are:

- 1) Neither the specific cell type(s) nor the specific MMP subtype(s) responsible for TRPML1 regulation of fibrogenesis is resolved. At least three cell types (pmLF, IM Φ , AM Φ) and five MMP subtypes (MMP2, 8, 9, 12, 19) were identified. However, do they all contribute equally to fibrogenesis or one or a few of them exert more dominant roles? For cell type specificity, ideally, conditional knockout mice should be used. If these mice are not available, cell-type-specific rescue experiments using the global knockout mice should at least be tried to distinguish the contributions between fibroblasts and M Φ 's. For the MMPs, it would be unrealistic to test mice with each MMP knocked out individually, but at a minimum, they should be examined in primary cultures for TRPML1-dependent degradation of ECM using a standard matrix degradation assay following RNAi knockdown of individual MMPs.
- 2) For the MMPs, it appears that the mRNA expression levels (for MMP2, MMP9, MMP12, and MMP19, based on Fig. S3) were not changed. This dataset is rather incomplete as no data is shown for MMP8 or for MMP mRNAs in AM Φ cells. More importantly, the total protein expression levels for these MMPs in the three cell types were not reported. If the main point is the secretion rather than synthesis or turnover of these MMPs that are regulated by TRPML1, it would make sense to also include results on total protein levels rather than just the extracellular levels.
- 3) No explanation is given for the lack of additive effect between Trpml1 knockout and bleomycin treatment. Does this suggest overlapping mechanisms or perhaps the exacerbating effect of Trpml1 deletion should be examined at an earlier day following the drug treatment or with the use of a lower drug dose? It would also make sense to test the heterozygotes following bleomycin treatment for shorter days and/or at a lower dose.
- 4) It would be better if the scRNAseq analysis also included Trpml1^{-/-} samples. This might reveal the homeostatic changes in the lung tissue in the absence of TRPML1, further illustrating the pathways affected by this channel.
- 5) Although developing new TRPML1 agonists and testing their effects on MMP secretion are interesting and of potential therapeutic value, the results presented in the manuscript do not provide any evidence for any benefits with respect to pulmonary fibrosis that may be brought about by any of the small-molecule activators. In addition, this part of the work was brought in rather abruptly without an appropriate introduction. The two sentences in line 289 do not appear to be related.

Minor points:

- 1) Results section beginning with "Screen for inflammatory mediators, TIMPs and MMPs...." is hard to read. It is written like a laundry list, with no emphasis on what the most important findings are. This section should be better streamlined. The very long paragraph should be split into multiple short paragraphs. If total mRNA and protein expression data are available and show no changes between WT and Trpml1^{-/-}, they should be emphasized to help strengthen the point that TRPML1 primarily regulates the secretion rather than the expression of the MMPs.
- 2) Figure S6a shows that LC3 II levels in fed (DMSO) is markedly higher in Trpml1^{-/-} than WT cells. This may suggest a decreased autophagic flux under basal conditions. If so, then quantification based on normalization to this control can be misleading.
- 3) There is no mention of MMP9 in the first paragraph of the introduction. Is this MMP not previously known to be involved in pulmonary or other types of fibrosis? If so, it may be worth mentioning in the Results section that this is the first time MMP9 is implicated in pulmonary fibrosis (however, based on major point #1, whether MMP9 deficiency actually contributes to fibrosis has not been established).
- 4) "primewave-8 perturbation" should be explained.
- 5) Line 175-178, "The effect of Bleo treatment in Trpml1^{-/-} mice did not result in a further exacerbation and was not different from PBS-treated Trpml1^{-/-} mice as well as Bleo treated WT mice, while being different from PBS-treated WT mice ", remove "The effect of" and correct "form" to "from".
- 6) Line 279-282, authors may want to add "(see Discussion)" after the sentence beginning with "Specifically, the finding that MMP8 and MMP12 were reduced is well in line...." Without reading the Discussion, it is very difficult to understand why they are "well in line".
- 7) Line 289, "Next to known" is not a standard English phrase.
- 8) Fig. 6c, what are the purple dots in the cartoon?
- 9) Fig. 6f, based on the numbers shown, it is hard to believe that WR01-001 30 μ M had no significant effect on Trpml1^{-/-} cells.
- 10) Fig. 6h, what is the point showing the results from THP-1 cells if the effects of the drugs were so much weaker than those in primary macrophages?
- 11) Fig. 7a-b, does 1 μ M WR1-002 affect MMP12 levels in SN of Trpml1^{-/-} cells? This experiment should also be carried out for other MMPs, including a negative control, i.e., an MMP of which the secretion is independent of TRPML1.

12) In the Discussion, all mentions of MMP levels should emphasize that they are extracellular levels.

Referee #1:

The study focuses on molecular mechanisms of pulmonary fibrosis. It identifies TRPML1 as a regulator of MMP release, thus providing a **novel link between lysosomal biology and ECM turnover**. The findings might have potential implications for understanding pulmonary fibrosis pathogenesis and for developing TRPML1-targeted therapies. The **concept is innovative** and could advance the field of fibrosis research. However, further validation is necessary to firmly establish causality and therapeutic relevance.

A: We thank the Reviewer for their effort to evaluate our MS and for their opinion that “The concept is innovative and could advance the field of fibrosis research.”

Major Concerns:

- Introduction: The introduction would benefit from expansion to include a more comprehensive discussion of the role of ECM components and individual MMPs in lung fibrosis. Furthermore, the involvement of different lung-resident cell types in fibrosis progression should be described. A more complete summary of TRPML1 biology is also warranted to provide sufficient context for its proposed role in disease.

A: This is a fair point and we have modified the Introduction accordingly addressing all points listed.

- Figure 1: Quantification of TRPML1 is missing. Western blotting, qPCR, or immunostaining should be included to demonstrate differences in TRPML1 expression across experimental conditions. This evidence is critical for supporting the proposed mechanistic role of TRPML1.

A: The Reviewer asks for expression data for TRPML1. In Fig.3 we provide qPCR data and even direct TRPML1 current measurements using endolysosomal patch-clamp for different WT vs. KO. In Fig. 1 we did not use other experimental conditions as basal for WT and KO cells. We would like to emphasize that this TRPML1 KO model was obtained from Prof. Dr. Susan Slaugenhaupt, Harvard University and has been used and assessed in numerous studies before.

- Figure 2: The choice of a 10-day time point for fibrosis assessment requires justification, as fibrotic remodelling generally occurs at later stages (14-28 days). The rationale should be clearly stated, and if available, data from additional time points should be included to substantiate the conclusions.

A: We initially aimed to perform Bleo treatment for 14 days; however, too many mice of our approved cohort reached a clinical score of 2–3, presenting with weight loss and reduced activity. Importantly, these symptoms occurred in both WT and TRPML1 KO mice without noticeable differences between the groups and is not uncommon (see e.g., Cowley et al. 2019 doi: 10.30802/AALAS-CM-18-000060). According to the animal welfare regulations in Germany (TierSchG and Tierschutz-Versuchstierverordnung) as well as the guidelines of our local animal ethics committee, animals must be euthanized once they reach defined humane endpoints. For this reason, extending the treatment to 14 days was not ethically permissible. Importantly, the time point of 10 days coincides with the well-established transition from

the inflammatory to the fibrotic phase (Moeller et al. 2008; Chaudhary et al. 2006), when fibrogenesis is already underway. In fact, in the WT mice, significant accumulation of ECM components was observed already at this early time point due to the treatment. We have clarified this Point in the revised MS and also emphasized why we performed the Bleo study over 10 days only.

We would like to point out that the fibrosis phenotype of TRPML1 KO is already shown in Fig.1 while Fig.2 is an additional dataset that, in our opinion is optional for this story. Fig. 2 shows that at day 10 TRPML1 without Bleo has the same fibrosis level as WT with Bleo or TRPML1 KO with Bleo. We have further clarified this point in the respective sections and explained why we did not perform longer term exp. We hope that this addresses the concerns sufficiently. Alternatively, we can also omit all Bleo data from the MS as they are not necessary for the claim that TRPML1 KO shows a fibrosis phenotype at basal levels.

In addition and albeit not asked for by the Reviewers we now also show desmosine levels (ELISA) in Fig.1 to further corroborate the observed Verhoeff-Van Gieson phenotype. See new Fig.1 J (new experiment).

- Cell type specificity: It remains unclear whether TRPML1-mediated regulation of MMPs is restricted to macrophages or whether other lung cell types contribute. This point should be clarified through either additional data or an **expanded discussion**.

A: We have focused throughout the MS on the fact that the secretion of several MMPs from IM, AM and fibroblasts is affected simultaneously in the TRPML1 KO. Specifically, we found MMP2 and 19 being reduced in SN of fibroblasts, MMP9 being reduced in SN of IM and fibroblasts, MMP8 and 12 being reduced in SN of AM. Hence, we would argue that secretion not only from diff. macrophages but also from fibroblasts does contribute. We would further argue that it is not the reduction or loss of a single MMP that causes the phenotype or is dominant (most MMPs are able to degrade collagen and would be expected to at least partially compensate for each other) but rather the fact that several MMPs are affected at the same time. We have modified the Discussion accordingly to further clarify this point. However, in addition and to further support this hypothesis we have now also performed single KD experiments. We have successfully knocked down single MMPs in WT fibroblasts and assessed their impact on collagen degradation using a **zymography assay**. As a result, we found that all tested MMPs were after knockdown having similar effects on collagen in the zymography assay, underscoring similar relevance of these MMPs rather than demonstrating dominance of a single MMP (new experiments).

Minor concerns:

- Figure set-up and formatting:
 - All figures should be presented in a uniform "letter" format.

A: Unfortunately, letter format is not possible for all figures. We nevertheless hope that the Reviewer might find our current layout and presentation of the data acceptable.

- Fonts should be consistent across figures, with sufficient size for readability.

A: We have checked and adapted the fonts as requested

- Panel ordering should strictly follow alphabetical order in all figures.

A: We have checked and adapted the panel ordering accordingly

- In Figure 1e-f, the SR quantification should be presented after the corresponding staining images.

A: Thanks for this comment. We agree. This has been changed now.

- The designation Trpml1-/- should be italicized.

A: We agree. This has been changed.

- Magnification panels should align precisely with the areas indicated in the corresponding lung panels (e.g., Figure 1).

A: We spotted the error in Fig.1c and have corrected it accordingly; we have also checked this throughout the MS.

- Figure 2: the colors in the legends do not match the staining panels. Consistent labeling should be used.

A: Thanks for drawing our attention to this. We have changed this accordingly wherever necessary.

- Figure 4: This figure shows no significant changes in inflammatory mediators and TIMPs, may be better suited for the supplementary section.

A: Correct. This Fig. may indeed be well suited for the Suppl. However, the Suppl. Material = Expanded View in EMBO J. is limited to 5 Fig. only and we feel that this dataset still provides relevant information even if "negative" data, as fibrosis is often associated with inflammation and changes in inflammatory mediators, which surprisingly was not the case for TRPML1KO. Hence, we feel this info is important enough to remain in the main part of the MS but if the Reviewer insists we can still move it to Expanded View and omit another Fig. currently in EV and move that one to "Appendix".

- Supplementary material: In Figure S2, a summary table of MMP expression patterns would enhance clarity and allow comparison across datasets.

A: This is a good idea. We are happy to also present the results of MMP expression in Table format. We leave it up to the Reviewer(s) and Editor to make a final decision on whether below Table may be part of the final MS or for the Reviewers' information only.

	Expression RNASeq AM Φ	Expression RNASeq IM Φ	Expression RNASeq pmLF	Effect
MMP2	-	-	+++	Reduced in ML1KO
MMP3	-	-	+++	Not changed in ML1KO
MMP7	-	-	-	Not tested
MMP8	+	-	-	Reduced in ML1KO
MMP9	-	++	+	Reduced in ML1KO
MMP10	-	-	-	Not tested
MMP11	-	-	+	Not tested
MMP12	++	++	-	Reduced in ML1KO
MMP13	-	+	-	Not changed in ML1KO
MMP14	+	+	+++	Not changed in ML1KO
MMP15	-	-	-	Not tested
MMP16	-	-	-	Not tested
MMP17	-	-	-	Not tested
MMP19	++	+	+++	Reduced in ML1KO
MMP20	-	-	-	Not tested
MMP21	-	-	-	Not tested
MMP23	-	-	+	Not tested
MMP24	-	-	-	Not tested
MMP25	-	-	-	Not tested
MMP27	-	-	(+)	Not tested
MMP28	-	-	-	Not tested

Referee #2:

TRPML1 is a widely expressed Ca²⁺-permeable cation channel localized in the late endosomes and lysosomes. This channel has been implicated in many physiological functions and diseases, most notably mucopolidosis type IV, a lysosomal storage disease caused by mutations in the TRPML1 gene. In the submitted study, by examining the effect of Trpml1 gene ablation in the mouse lung, **the authors noticed an obvious pulmonary fibrosis phenotype**, which was not further exacerbated by the treatment of bleomycin. This **interesting observation** led to further interrogation of the underlying cause of lung injury due to TRPML1 deficiency. By analyzing the expression and secretion of matrix metalloproteinases (MMPs) in TRPML1-expressing cells present in the lung, the authors concluded that TRPML1 regulates pulmonary fibrosis by promoting exocytosis of certain MMP subtypes from primary murine lung fibroblasts (pmLFs), and interstitial and alveolar macrophages (IMΦ and AMΦ). These MMPs then degrade extracellular matrix (ECM) to counter the fibrogenic process. These findings are **very interesting**, as they reveal a **novel aspect of TRPML1 function in fibrogenesis**, with potential mechanistic insights. **The analysis of the pulmonary fibrosis phenotypes was thorough**; the single-cell RNA-seq analysis was informative; the extensive evaluation of large numbers of inflammatory cytokines, MMPs, and TIMPs in the tissue and supernatants of cultured cells was commendable. However, despite the large amount of work, the mechanistic insights revealed based on the data shown in the manuscript are rather incomplete. The major issues are:

A: We thank the Reviewer for their positive evaluation of our MS and for their opinion that our findings are “**very interesting**” and “**novel**”. We have followed the Reviewer’s advice/suggestions regarding additional exp. to improve and further buttress our claims. See detailed responses below.

1) Neither the specific cell type(s) nor the specific MMP subtype(s) responsible for TRPML1 regulation of fibrogenesis is resolved. At least three cell types (pmLF, IMΦ, AMΦ) and five MMP subtypes (MMP2, 8, 9, 12, 19) were identified. However, do they all contribute equally to fibrogenesis or one or a few of them exert more dominant roles?

A: We thank the Reviewer for this comment. As concluded in the Results/Discussion not all MMPs are expressed equally by the different cell types assessed.

Diff. macrophages express distinct subsets of MMPs and likewise fibroblasts express a specific subset of MMPs. Interestingly, not all MMPs are affected by loss of TRPML1. Therefore, we concluded that not all MMPs are contributing but those MMPs which are changed in the KO likely contribute all in combination to the overall effect. Strikingly, all affected MMPs are reduced and none was found to be increased. This is also remarkable given that TRPML3 KO showed the opposite effect i.e., increased MMP levels (Spix et al., Nature Commun., 2022) and the opposite lung function phenotype. To address the Reviewer’s question experimentally, we performed siRNA experiments for several MMPs to assess their single effects on collagen degradation. **See further comments and the results of the requested KD experiments on this topic below.**

For cell type specificity, ideally, conditional knockout mice should be used. If these mice are not available, cell-type-specific rescue experiments using the global knockout mice should at least be tried to distinguish the contributions between fibroblasts and MΦ's.

A: Conditional TRPML1 KO mice are unfortunately not available. Alternatively, we have now done KD experiments to assess the impact of different MMPs. In addition, for the different cell types involved we have performed experiments using TRPML1 agonist to assess the extent of increased levels of selected MMPs. See new Fig.7. KO controls and a control MMP which was unaffected in KO, MMP3 are shown as well (new experiments, see also next Point, Point 5 and Minor Points 9 and 11 below).

For the MMPs, it would be unrealistic to test mice with each MMP knocked out individually, but at a minimum, they should be examined in primary cultures for TRPML1-dependent degradation of ECM using a standard matrix degradation assay following RNAi knockdown of individual MMPs.

A: In order to evaluate the impact of diff. MMPs on matrix/collagen degradation, experiments as suggested above with MMP KO mice could theoretically be performed. For several MMPs commercial KO mice are available. Lungs could be isolated and stained for collagen as shown in Fig.1 and 2 for WT and MMP KO. But as the Reviewer points out it is indeed “unrealistic to test mice with each MMP knocked out individually“ last but not least due to the breeding times after purchasing such KO models. In addition, complete KO of MMPs would not correspond to the partial reductions in MMPs that we see in TRPML1KO. Hence, we followed the Reviewer’s alternative suggestion, to do cell culture exp. with siRNA, and using a matrix degradation assay: <https://www.abcam.com/en-us/products/assay-kits/collagen-degradation-zymography-assay-kit-fluorometric-ab234624> Using this assay we have tested now several MMPs in KD exp. As a result, we found that all tested MMPs were, after knockdown having similar effects on collagen in the zymography assay, underscoring similar relevance of these MMPs rather than demonstrating dominance of a single MMP (new experiments). We have also expanded the Discussion to explain this aspect.

2) For the MMPs, it appears that the mRNA expression levels (for MMP2, MMP9, MMP12, and MMP19, based on Fig. S3) were not changed. This dataset is rather incomplete as no data is shown for MMP8 or for MMP mRNAs in **AMΦ cells**. More importantly, the total protein expression levels for these MMPs in the three cell types were not reported. If the main point is the secretion rather than synthesis or turnover of these MMPs that are regulated by TRPML1, it would make sense to also include results on total protein levels rather than just the extracellular levels.

A: We thank the Reviewer for this comment. We completed the qPCR data accordingly and where appropriate antibodies were available performed WB analysis of MMP protein levels to confirm qPCR data (**new experiments**). Neither in qPCR nor in WB significant diff. in expression was found. See new EV2.

3) No explanation is given for the lack of additive effect between Trpml1 knockout and bleomycin treatment. Does this suggest overlapping mechanisms or perhaps the exacerbating effect of Trpml1 deletion should be examined at an earlier day following the drug treatment or with the use of a lower drug dose? It would also make sense to test the heterozygotes following bleomycin treatment for shorter days and/or at a lower dose.

A: This is a good point. The Bleo dataset is an additional dataset that we believe is not strictly necessary for the overall conclusion that TRPML1 KO shows a fibrosis phenotype in the lung. Fig. 1 would be sufficient to underscore this claim.

Fig.2 provides additional information in the sense that the extent of fibrosis seen in TRPML1 KO is at least as high as effects seen with Bleo in WT at day 10, providing a comparison with an **established fibrosis inductor**.

We initially aimed to perform Bleo treatment for 14 days; however, too many mice of our approved cohort reached a clinical score of 2–3, presenting with weight loss and reduced activity. Importantly, these symptoms occurred in both WT and TRPML1 KO mice without noticeable differences between the groups (see also Cowley et al. 2019 doi: 10.30802/AALAS-CM-18-000060). According to the animal welfare regulations in Germany (TierSchG and Tierschutz-Versuchstierverordnung) as well as the guidelines of our local animal ethics committee, animals must be euthanized once they reach defined humane endpoints. For this reason, extending the treatment to 14 days was not ethically permissible. Importantly, the time point of 10 days coincides with the well-established transition from the inflammatory to the fibrotic phase (Moeller et al. 2008; Chaudhary et al. 2006), when fibrogenesis is already underway. In fact, in the WT mice, significant accumulation of ECM components was observed already at this early time point due to the treatment. We have added this information to the MS.

Earlier time points of Bleo treatment or lower doses we have tried before but Bleo did not evoke visible phenotypes under those conditions. Since TRPML1 KO mice show already a strong basal effect at time point zero similar to Bleo treatment for 10 days in WT we are unsure if lower doses or earlier timepoints of Bleo treatment could possibly show an even stronger phenotype at an earlier timepoint in TRPML1 KO. More problematic, we currently have no permission to do additional doses or time points; we would need to reapply for this with the legal authorities in Germany and currently these applications take up to one year. We have now clarified this point in the respective Results/Discussion sections, explaining also as to why we did not perform longer term exp. Alternatively, we can also omit all Bleo data from the MS if the Reviewer prefers as they are in our opinion not necessary for the claim that TRPML1 KO shows a fibrosis phenotype at basal levels.

In addition and albeit not asked for by the Reviewers we now also show **desmosine** levels (ELISA) in Fig.1 to further corroborate the observed phenotype and specifically the Verhoeff van Gieson results. See new Fig.1 J (**new experiment**).

4) It would be better if the scRNAseq analysis also included Trpml1^{-/-} samples. This might reveal the homeostatic changes in the lung tissue in the absence of TRPML1, further illustrating the pathways affected by this channel.

A: A similar request was made in Point 2. Regarding MMPs, we now provide more qPCR and WB data of WT vs. TRPML1 KO accordingly (**new experiments**). See EV2.

5) Although developing new TRPML1 agonists and testing their effects on MMP secretion are **interesting** and of **potential therapeutic value**, the results presented in the manuscript do not provide any evidence for any benefits with respect to pulmonary fibrosis that may be brought about by any of the small-molecule activators. In addition, this part of the work was brought in rather abruptly without an appropriate introduction. The two sentences in line 289 do not appear to be related.

A: We agree with the Reviewer. We now provide effects of TRPML1 agonist, in addition to MMP12 in Fig.7 for other MMPs, MMP2, 9, and 19, all with the same result: The TRPML1 agonist increases MMP levels in WT but not in TRPML1KO (**new experiments, see also Minor Points 9 and 11 below**). We argue based on these results that increasing those MMP levels with TRPML1 agonist would enable more collagen/elastin degradation and hence potentially improve lung fibrosis in patients. We agree it would be ideal to additionally test TRPML1 agonists in murine or other animal fibrosis models in vivo but we think, given the long time it takes to get approval for such experiments that those are experiments that would be more suitable for an independent follow up study.

Minor points:

1) Results section beginning with "Screen for inflammatory mediators, TIMPs and MMPs...." is hard to read. It is written like a laundry list, with no emphasis on what the most important findings are. This section should be better streamlined. The very long paragraph should be split into multiple short paragraphs. If total mRNA and protein expression data are available and show no changes between WT and *Trpml1*^{-/-}, they should be emphasized to help strengthen the point that TRPML1 primarily regulates the secretion rather than the expression of the MMPs.

A: We agree with the Reviewer and have modified the description of the multiple results accordingly. The amount of tested analytes and MMPs and their different expression levels and expression patterns in the lung unfortunately make this section complicated to embed in an easy read manner. We tried nevertheless to rephrase and modify this section where possible in the hope that the Reviewer will now find it easier to follow. We have tried also to split the paragraph into multiple short paragraphs as suggested. Nevertheless, we deem these results all relevant as we assessed many factors potentially playing a role for the

observed fibrosis phenotype allowing us to extract eventually, as a result of these experiments only a few MMPs as relevant (only **5 out of >20**).

2) Figure S6a shows that LC3 II levels in fed (DMSO) is markedly higher in *Trpml1*^{-/-} than WT cells. This may suggest a decreased autophagic flux under basal conditions. If so, then quantification based on normalization to this control can be misleading.

A: Indeed, autophagy is reduced in TRPML1 KO as reported in several previous publications. Importantly, we would like to argue here that despite those differences in autophagy, intracellular collagen levels and/or degradation were not affected.

It is important to note that the presented dataset was designed to assess autophagic flux rather than basal autophagy. Basal LC3-II levels under fed+DMSO conditions alone cannot be used to draw conclusions on autophagic flux. The LC3-II level reflects a steady-state balance between autophagosome formation and degradation, hence an increased basal level may result from either enhanced formation or reduced clearance. Therefore, flux must be assessed by comparing basal conditions to induced autophagy (e.g., starvation in HBSS). Bafilomycin further unmask this difference by blocking degradation and revealing the turnover pool.

If all data are normalized to WT fed+DMSO, the relatively higher basal LC3-II in KO can mask the true flux. For this reason, we analyzed flux within each genotype separately: Normalizing within the genotypes and then comparing basal to starvation levels. Using this strategy, WT cells show robust flux, while TRPML1 KO cells display reduced flux, consistent with published reports of impaired autophagy in this genotype. To make this point clearer to Reviewer and Readers we may present the data as shown below and it would be up to the Reviewer to decide whether this presentation makes it clearer:

3) There is no mention of MMP9 in the first paragraph of the introduction. Is this MMP not previously known to be involved in pulmonary or other types of fibrosis? If so, it may be worth mentioning in the Results section that this is the first time MMP9 is implicated in pulmonary fibrosis (however, based on major point #1, whether MMP9 deficiency actually contributes to fibrosis has not been established).

A: The Reviewer is right. We have mentioned MMP9 now also in the Introduction. Indeed, currently we only mention MMP9 in the Discussion. The corresponding phrase has now been moved, in part to the Introduction: *Additionally, albeit also controversially discussed*⁶¹, *in the lung evidence for MMP9 suggests potential antifibrotic roles*⁶², (Discussion) *with MMP9 (collagenase 4) reportedly being capable of degrading several collagens including collagen I, III, IV, V, X, XI and XIV*⁶³⁻⁶⁸ (Introduction).

4) "primewave-8 perturbation" should be explained.

A: We agree. An explanation is now provided in the text and Ref. are provided for further explanations (Spix et al., 2022 and Vanoirbeek et al., 2010)

5) Line 175-178, "The effect of Bleo treatment in Trpml1^{-/-} mice did not result in a further exacerbation and was not different from PBS-treated Trpml1^{-/-} mice as well as Bleo treated WT mice, while being different from PBS-treated WT mice ", remove "The effect of" and correct "form" to "from".

A: This has been corrected. Many thanks for spotting this typo.

6) Line 279-282, authors may want to add "(see Discussion)" after the sentence beginning with "Specifically, the finding that MMP8 and MMP12 were reduced is well in line...." Without reading the Discussion, it is very difficult to understand why they are "well in line".

A: We have deleted this sentence from the Results section and now discuss TRPML1KO results in comparison to TRPML3KO results in the Discussion only.

7) Line 289, "Next to known" is not a standard English phrase.

A: We agree. This phrase has been removed.

8) Fig. 6c, what are the purple dots in the cartoon?

A: The purple dots shall represent "lysosomal luminal content", according to our hypothesis including also MMPs. We have specified this in the Fig. legend accordingly.

9) Fig. 6f, based on the numbers shown, it is hard to believe WR01-001 that 30 μ M had no significant effect on Trpml1^{-/-} cells.

A: We double checked this and confirmed that the difference is "not significant" when applying Two Way ANOVA with multiple comparison on DMSO (see reassessment shown in Fig. below). We would like to draw the Reviewer's attention to the fact that in our MMP experiments in Fig.7a we used WR01-002 as this is the compound, we moved forward with based on the functional evaluation provided in Fig.6. WR01-001 is not a very potent compound and was not considered further. We have now tested WR01-002 in addition also in TRPML1 KO cells using MMP ELISA to make sure there is no effect in the KO (**new experiments**). See also point below. Importantly, in WT cells TRPML1 agonist increases MMP levels while it has no effect in KO. This is an important finding as now we show not only reduced MMP levels in TRPML1KO but also that activation of TRPML1 increases the SN levels of the exact same MMPs in WT while being inactive in the KO.

10) Fig. 6h, what is the point showing the results from THP-1 cells if the effects of the drugs were so much weaker than those in primary macrophages?

A: The idea here was to show effects also in human macrophages, but we agree these data can be omitted if the Reviewer believes they are not useful.

11) Fig. 7a-b, does 1 µM WR1-002 affect MMP12 levels in SN of Trpm1^{-/-} cells? This experiment should also be carried out for other MMPs, including a negative control, i.e., an MMP of which the secretion is independent of TRPML1.

A: We absolutely agree with the Reviewer. Corresponding experiments including TRPML1KO for MMP12 and other MMPs were performed and added accordingly to Fig.7 (**new experiments**).

12) In the Discussion, all mentions of MMP levels should emphasize that they are extracellular levels.

A: We have added “extracellular” in the Discussion.

Dear Dr. Grimm,

Thank you again for submitting your amended manuscript (EMBOJ-2025-121886R1) to The EMBO Journal. Please accept my apologies for the unusual protraction with the reassessment due to delayed expert input at this time of the year and detailed discussion in the editorial team. Your revised study was sent back to the referees for their scientific reassessment, and we have received detailed re-reports from both of them, which I enclose below. As you will see, the referees state that the work has been substantially enhanced by the revisions and they are now in favour of publication, pending minor revision.

Thus, we are pleased to inform you that your manuscript has been accepted in principle for publication in The EMBO Journal.

Please carefully consider the remaining minor issues still raised by referee #2 by adjusting data presentation and discussion of the findings in the manuscript text.

Also, we now need you to take care of a number of minor issues related to formatting and data annotation, which I will share shortly in a separate message, together with additional changes and requests by our production team for Source Data provision.

As you might have seen on our web page, every paper at the EMBO Journal now includes a 'Synopsis', displayed on the html and freely accessible to all readers. The synopsis includes a 'model' figure as well as 2-5 one-short-sentence bullet points that summarize the article. I would appreciate if you could provide this figure and the bullet points.

Please submit a revised version of the manuscript using the link enclosed below, addressing the advisor's comments.

Thank you again for giving us the chance to consider your manuscript for The EMBO Journal, I look forward to hearing from you and receiving your final revised version of the manuscript.

Kind regards,

Daniel Klimmeck

>> Please limit the keywords for your study to maximally five.

>> Author Contributions: Remove the author contributions information from the manuscript text. Note that CRediT has replaced the traditional author contributions section as of now because it offers a systematic machine-readable author contributions format that allows for more effective research assessment. and use the free text boxes beneath each contributing author's name to add specific details on the author's contribution.

More information is available in our guide to authors.
<https://link.springer.com/journal/44318/submission-guidelines>

>> Section order should be as follows: title page with complete author information, abstract, keywords, introduction, results, discussion, methods, data availability section, acknowledgements, disclosure and competing interests statement, references, main figure legends, tables, expanded figure legends.

>> Adjust the title of the 'Competing Financial Interests' section to 'Disclosure and Competing Interests Statement'. Please add biotech company affiliations of J. G.-F. and L. O. M. .

>> Figure callouts: there is a callout for a Fig. S1, please correct.

>> Funding: Please add the complete list of funders to our system; the list will be linked to the central database in PubMed in all published articles, it is therefore essential that the list in our system is complete and accurate.

>> Add a Reagents and Tools table to the Methods section, as a separate file using the existing template in the Guide For Authors, listing key reagents, experimental models, software and relevant equipment.

>> Add a separate 'Statistical Analysis' section to the Methods part, detailing the algorithms and statistical tests applied.

>> Author Checklist: responses need to be selected in column D.

>> Please recheck bioRxiv entry Gote-Schniering et al. (2025) and include as PREPRINT citation in the reference list in case it was not published as regular article in the meantime.

>> Consider additional changes and comments from our production team as indicated below:

- Figure legends:

1. Please define the annotated p values ****/***/**/* as well as provide the exact p-values for the same in the legend of figure EV3 B as appropriate.
2. Please note that the exact p values are not provided in the legends of figures 1B, D, F, H, I, J; 2B, D, F, H; 3A, F; 4C, 5H-N; 6B, E, F, H; 7A, EV1 B, EV3 A, EV5 B
3. Please note that information related to n is missing in the legend of figure EV4 E
4. Please note that the error bars are not defined in the legend of figure EV4 E

Further information is available in our Guide For Authors: <https://link.springer.com/journal/44318/submission-guidelines>

Referee #1:

The authors carefully and thoroughly addressed the comments. The requested quantifications were performed. The requested justifications were provided as well, and Introduction and Discussion sections modified accordingly. I have no further concerns.

Referee #2:

TRPML1 is a widely expressed Ca²⁺-permeable cation channel localized in the late endosomes and lysosomes. This channel has been implicated in many physiological functions and diseases, most notably mucopolidosis type IV, a lysosomal storage disease caused by mutations in the TRPML1 gene. In the revised ms, the authors provide additional experimental evidence that TRPML1 is protective against pulmonary fibrosis by promoting exocytosis of several matrix metalloproteinases (MMPs) from lung fibroblasts, and interstitial and alveolar macrophages. These MMPs then degrade extracellular matrix (ECM) to counter the fibrogenic process. These interesting findings reveal a novel aspect of TRPML1 function in fibrogenesis, with potential mechanistic insights. The new data further clarified the specific MMP subtypes and cell types regulated by TRPML1 and demonstrated the specificity of the newly developed TRPML1 agonist in MMP secretion. Overall, the ms is much improved.

The authors asked for the reviewer's opinion on some of the data. I would suggest removing the THP-1 cell data in Fig. 6H. For Fig. 5EV_B, the version shown in the rebuttal letter looks better than the original one.

The authors addressed the remaining editorial issues.

Dear Dr. Grimm,

Thank you for submitting the revised version of your manuscript. I have now evaluated your amended manuscript and concluded that the remaining minor concerns have been sufficiently addressed.

I am thus pleased to inform you that your manuscript has been accepted for publication in the EMBO Journal.

Best regards,

Daniel Klimmeck

Daniel Klimmeck, PhD
Senior Editor
The EMBO Journal
EMBO
Postfach 1022-40
Meyerhofstrasse 1
D-69117 Heidelberg
contact@embojournal.org

Please note that it is The EMBO Journal policy for the transcript of the editorial process (containing referee reports and your response letters) to be published as an online supplement to each paper. If you should prefer removal of any referee-only figures included in the point-by-point response(s), e.g. because they may still be used for future publication or because they have been reproduced from published work by others, please do let us know immediately via response email.

More information is available here: <https://link.springer.com/partners/embo-press/editorial-policies#Peer%20review>